# ReLU soothes NTK conditioning and accelerates optimization for wide neural networks

## Abstract

Non-linear activation functions are well known to improve the expressivity of neural networks, which is the main reason of their wide implementation in neural networks. In this work, we showcase a new and interesting property of certain non-linear activations, focusing on the most popular example of its kind – Rectified Linear Unit (ReLU). By comparing the cases with and without this non-linear activation, we show that the ReLU has the following effects: (a) *better data separation*, i.e., a larger angle separation for similar data in the feature space of model gradient, and (b) *better NTK conditioning*, i.e., a smaller condition number of neural tangent kernel (NTK). Furthermore, we show that the ReLU network depth (i.e., with more ReLU activation operations) further magnifies these effects. Note that, without the non-linear activation, i.e., in a linear neural network, the data separation and NTK condition number always remain the same as in the case of a linear model, regardless of the network depth. Our results imply that ReLU activation, as well as the depth of ReLU network, helps improve the worst-case convergence rate of GD, which is closely related to the NTK condition number.

## 1 Introduction

Non-linear activation functions, such as rectified linear unit (ReLU), are well known for their ability to increase the expressivity of neural networks. A non-linearly activated neural network can approximate any continuous function to arbitrary precision, as long as there are enough neurons in the hidden layers (Hornik et al., 1989; Cybenko, 1989; Hanin & Sellke, 2017), while its linear counterpart – linear neural network, which has no non-linear activation functions applied, can only represent linear functions of the input. In addition, deeper neural networks, which have more non-linearly activated layers, have exponentially greater expressivity than shallower ones (Telgarsky, 2015; Poole et al., 2016; Raghu et al., 2017; Montufar et al., 2014; Wang et al., 2018), indicating that the network depth promotes the power of non-linear activation functions.

In this paper, we showcase a new and interesting property of certain non-linear activations, focusing on the ReLU instance: ReLU improves data separation in the feature space of model gradient, and helps to decrease the condition number of neural tangent kernel (NTK). We also show that the depth of the ReLU network further magnifies these effects, namely, a deeper ReLU activated neural network has a better data separation and a smaller NTK condition number, than a shallower one.

Specifically, we first show the *better separation phenomenon*, i.e., the improved data separation for similar data in the model gradient feature space. We prove that, for an infinitely wide ReLU network $f$ at its random initialization, any pair of data input vectors $\mathbf{x}$ and $\mathbf{z}$ that have similar directions (i.e., small but non-zero angle $\theta_{in}$ between $\mathbf{x}$ and $\mathbf{z}$) become more directionally separated in the model gradient space (i.e., model gradient angle $\phi$ between $\nabla f(\mathbf{x})$ and $\nabla f(\mathbf{z})$ is larger than $\theta_{in}$). We also find that deeper ReLU networks result in even better data separation, i.e., larger $\phi$.

We further show the *better NTK conditioning* property of ReLU, i.e., smaller NTK condition number. First, we prove that, as a consequence of the better data separation, the NTK condition number of a infinitely wide ReLU network is strictly smaller than that of the Gram matrix, if the dataset contains two non-degenerate samples. Moreover, as the ReLU network depth increases, the NTK condition number monotonically decreases. Then, we remove this data size assumption on two-layer ReLU networks, and prove the same better NTK conditioning, regardless of the data size as long as the dataset is not degenerated. The intuition is that, if there exists a pair of similar inputs $\mathbf{x}$ and $\mathbf{z}$ in the

training set (i.e., the angle between $\mathbf{x}$ and $\mathbf{z}$ is small), which is usually the case for large datasets, then the Gram matrix and NTK of linear neural networks must have close-to-zero smallest eigenvalues, resulting in extremely large NTK condition numbers. The ReLU activation make these similar data more separated (enlarges the small angles between data), hence it helps to increase the smallest eigenvalues of NTK, which in turn leads to a smaller NTK condition number.

Note that, when the non-linear activation is absent, as in an infinitely wide linear neural network $\bar{f}$ of any finite depth, the model gradient angle $\bar{\phi}$ is always equivalent to the input angle $\theta_{in}$, and the NTK condition number $\bar{\kappa}$ also remains identical to $\kappa_0$ of the Gram matrix. With this comparison, we conclude that the better separation phenomenon, i.e., $\phi > \theta_{in}$, and the better NTK conditioning, i.e., $\kappa < \kappa_0$, observed for ReLU networks, are attributed to the ReLU non-linear activation.

We experimentally verify these findings on finite but wide neural networks. It also suggests that these results hold for finite networks.

**Condition number and optimization theory.** Recent optimization theories showed that the NTK condition number, or the smallest eigenvalue of NTK, controls the theoretical convergence rate of gradient descent algorithms on wide neural networks (Du et al., 2018; 2019a; Liu et al., 2022). Combined with these theories, our findings imply that: (a), the ReLU activation function helps improving the worst-case convergence rate of gradient descent, and (b), deeper wide ReLU networks have faster convergence rate than shallower ones. Experimentally, we indeed find that deeper ReLU networks converges faster than shallower ones.

In this paper, we focus on the special case of ReLU, the most commonly used non-linear activation function. It remains theoretically an open question what are the effects of other non-linear activations on NTK conditioning and theoretical convergence rates. While it need different analysis techniques and we would like to leave it as a future work, we provide some preliminary numerical results in Appendix F. It suggests that the non-linear activation effect on the NTK conditioning can be positive (decreasing $\kappa$, as for *tanh*) or negative (increasing $\kappa$, as for *sigmoid*). It is worth to note that, in either case, a larger network depth, where more non-linear activation are operated, magnifies the effect.

**Contributions.** We summarize our contributions below. We find that:

- the ReLU activation function induces better separation between similar data in the feature space of model gradient. A larger depth of the ReLU network magnifies this better separation phenomenon.
- ReLU has the effect of decreasing the condition number of the NTK matrix. A larger depth of the ReLU network further enhances this better NTK conditioning property.
- This better NTK conditioning property leads to faster convergence rate of gradient descent. We empirically verify this on various real world datasets.

The paper is organized as follow: Section 2 describes the setting and defines the key quantities and concepts; Section 3 analyzes linear neural networks as the baseline for comparison; Section 4 and 5 discuss our main results on the better separation and better conditioning of ReLU non-linear activation, respectively; Section 6 discusses the implication on theoretical convergence rates; Section 7 concludes the paper. Proofs of theorems and main corollaries can be found in the appendix.

## 1.1 RELATED WORK

NTK and its spectrum have been extensively studied (Lee et al., 2019; Bietti & Mairal, 2019; Liu et al., 2020; Fan & Wang, 2020; Geifman et al., 2020; Nguyen et al., 2021; Belfer et al., 2021; Chen & Xu, 2021), since the discovery of constant NTK for infinitely wide neural networks (Jacot et al., 2018). For example, the NTK spectrum distribution of an infinitely wide neural network is shown to be similar to that of Laplace kernel (Geifman et al., 2020; Chen & Xu, 2021), and can be computed (Fan & Wang, 2020). Nguyen et al. (2021) analyzed the upper and lower bounds for the smallest NTK eigenvalue in $O()$ and $\Omega()$. Recent optimization theories on wide neural networks find connections between the theoretical convergence rate of gradient descent and the NTK condition number (Du et al., 2018; Liu et al., 2022). To the best of our knowledge, our work is the first to disclose the effect of a non-linear activation function on the NTK spectrum and condition number. We also note that those spectral analyses are in $O()$ and/or $\Omega()$, which absorbed certain constants. While this type of $O()/\Omega()$ analysis is fruitful on many topics, we realize that, in order to reveal the effect of the

non-linear activations, those constants should not be omitted and hence the above mentioned analysis does not apply to the problem we solve in this paper.

Studying a specific type of non-linear activation function, especially ReLU, is a common setting in the literature. This is largely due to the fact that ReLU has emerged to be the dominant choice of activation functions in neural networks used in practice, since Nair & Hinton (2010); Krizhevsky et al. (2012). ReLU activated neural networks have received wide research attention, ranging from optimization (Li & Yuan, 2017; Du et al., 2018; Zou et al., 2020), expressivity (Hanin & Sellke, 2017; Yarotsky, 2017; Wang et al., 2018), generalization (Zheng et al., 2019; Ji & Telgarsky, 2019; Cao & Gu, 2020), etc.

We are aware of a prior work (Arora et al., 2018) which has results of similar flavor. It shows that the depth of a linear neural network may help to accelerate optimization via an implicit pre-conditioning of gradient descent. We note that this prior work is in an orthogonal direction, as its analysis is based on the linear neural network, which is activation-free, while our work focus on the better-conditioning effect of ReLU activation function.

## 2 SETUP AND PRELIMINARIES

**Notations for general purpose.** We denote the set $\{1, 2, \cdots, n\}$ by $[n]$. We use bold lowercase letters, e.g., $\mathbf{v}$, to denote vectors, and capital letters, e.g., $A$, to denote matrices. Given a vector, $\|\cdot\|$ denotes its Euclidean norm. Inner product between two vectors is denoted by $\langle \cdot, \cdot \rangle$. Given a matrix $A$, we denote its $i$-th row by $A_{i:}$, its $j$-th column by $A_{:j}$, and its entry at $i$-th row and $j$-th column by $A_{ij}$. We also denote the expectation (over a distribution) of a variable by $\mathbb{E}[\cdot]$, and the probability of an event by $\mathbb{P}[\cdot]$. For a model $f(\mathbf{w}; \mathbf{x})$ which has parameters $\mathbf{w}$ and takes $\mathbf{x}$ as input, we use $\nabla f$ to denote its first derivative w.r.t. the parameters $\mathbf{w}$, i.e., $\nabla f := \partial f / \partial \mathbf{w}$.

**(Fully-connected) ReLU neural network.** Let $\mathbf{x} \in \mathbb{R}^d$ be the input, $m_l$ be the width (i.e., number of neurons) of the $l$-th layer, $W^{(l)} \in \mathbb{R}^{m_l \times m_{l-1}}$, $l \in [L+1]$, be the matrix of the parameters at layer $l$, and $\sigma(z) = \max\{0, z\}$ be the ReLU activation function. A (fully-connected) ReLU neural network $f$, with $L$ hidden layers, is defined as:

$$\alpha^{(0)}(\mathbf{x}) = \mathbf{x}$$
$$\alpha^{(l)}(\mathbf{x}) = \frac{\sqrt{2}}{\sqrt{m_l}} \sigma\left(W^{(l)} \alpha^{(l-1)}(\mathbf{x})\right), \quad \forall l \in \{1, 2, \cdots, L\}, \tag{1}$$
$$f(\mathbf{x}) = W^{(L+1)} \alpha^{(L)}(\mathbf{x}).$$

We also denote $\tilde{\alpha}^{(l)}(\mathbf{x}) \triangleq \frac{\sqrt{2}}{\sqrt{m_l}} W^{(l)} \alpha^{(l-1)}(\mathbf{x})$. Following the NTK initialization scheme (Jacot et al., 2018), these parameters are randomly initialized i.i.d. according to the normal distribution $\mathcal{N}(0, 1)$. The scaling factor $\sqrt{2}/\sqrt{m_l}$ is introduced to normalize the hidden neurons (Du et al., 2019b). We denote the collection of all the parameters by $\mathbf{w}$.

In this paper, we typically set the layer widths as

$$m_0 = d, \ m_{L+1} = 1, \ and \ m_l = m, \ for \ l \in [L]. \tag{2}$$

and call $m$ as the network width. We focus on the infinite network width limit, $m \to \infty$. We also define the network depth $L$ as the number of hidden layers.

**Linear neural network.** For a comparison purpose, we also consider a linear neural network $\bar{f}$, which is the same as the ReLU neural network $f$ (defined above), except that the activation function is the identity function $\sigma(z) = z$ and that the scaling factor is $1/\sqrt{m}$ (we adopt the network width setting in Eq.(2)):

$$\bar{\alpha}^{(0)}(\mathbf{x}) = \mathbf{x}, \ \bar{\alpha}^{(l)}(\mathbf{x}) = \frac{1}{\sqrt{m}} W^{(l)} \bar{\alpha}^{(l-1)}(\mathbf{x}), \ \forall l \in \{1, 2, \cdots, L\}, \ \bar{f}(\mathbf{x}) = W^{(L+1)} \bar{\alpha}^{(L)}(\mathbf{x}). \tag{3}$$

**Input feature and Gram matrix.** Given a dataset $\mathcal{D} = \{(\mathbf{x}_i, y_i)\}_{i=1}^n$, we denote its (input) feature matrix by $X$, where each row $X_{i:} = \mathbf{x}_i^T$. The Gram matrix is defined as $G = XX^T \in \mathbb{R}^{d \times d}$, with each $G_{ij} = \mathbf{x}_i^T \mathbf{x}_j$.

**Gradient feature and neural tangent kernel (NTK).** Given a model $f$ (e.g., a neural network) with parameters $\mathbf{w}$, we consider the vector $\nabla f(\mathbf{w}; \mathbf{x})$ is the gradient feature for the input $\mathbf{x}$. The NTK $\mathcal{K}$ is defined as

$$\mathcal{K}(\mathbf{w}; \mathbf{x}_1, \mathbf{x}_2) = \langle \nabla f(\mathbf{w}; \mathbf{x}_1), \nabla f(\mathbf{w}; \mathbf{x}_2) \rangle, \tag{4}$$

where $\mathbf{x}_1$ and $\mathbf{x}_2$ are two arbitrary network inputs. For a given dataset $\mathcal{D} = \{(\mathbf{x}_i, y_i)\}_{i=1}^n$, there is a gradient feature matrix $F$ such that each row $F_{i\cdot}(\mathbf{w}) = \nabla f(\mathbf{w}; \mathbf{x}_i)$ for all $i \in [n]$. The $n \times n$ NTK matrix $K(\mathbf{w})$ is defined such that its entry $K_{ij}(\mathbf{w})$, $i, j \in [n]$, is $\mathcal{K}(\mathbf{w}; \mathbf{x}_i, \mathbf{x}_j)$. It is easy to see that the NTK matrix

$$K(\mathbf{w}) = F(\mathbf{w})F(\mathbf{w})^T. \tag{5}$$

Note that the NTK for a linear model reduces to the Gram matrix $G$.

Recent discovery is that, when $m$ is sufficiently large or infinite, the NTK and gradient feature becomes almost constant during training by gradient descent (Jacot et al., 2018; Liu et al., 2020). Hence, it suffices to analyze these quantities only at the network initialization, which shall extend to all the optimization procedure.

**Condition number.** The *condition number* $\kappa$ of a positive definite matrix $A$ is defined as the ratio between its maximum eigenvalue and minimum eigenvalue:

$$\kappa = \lambda_{max}(A)/\lambda_{min}(A). \tag{6}$$

**Embedding angle and model gradient angle.** For a specific input $\mathbf{x}$, we call the vector $\alpha^{(l)}(\mathbf{x})$ as the $l$-embedding of $\mathbf{x}$. We also call $\nabla f$, i.e., the derivative of model $f$ with respect to all its parameters, as the model gradient. In the following analysis, we frequently use the following concepts: *embedding angle* and *model gradient angle*.

**Definition 2.1** (embedding angle and model gradient angle). *Given two arbitrary inputs* $\mathbf{x}, \mathbf{z} \in \mathbb{R}^d$, *define the l-embedding angle,* $\theta^{(l)}(\mathbf{x}, \mathbf{z}) \triangleq \arccos\left(\frac{\langle \alpha^{(l)}(\mathbf{x}), \alpha^{(l)}(\mathbf{z}) \rangle}{\|\alpha^{(l)}(\mathbf{x})\|\|\alpha^{(l)}(\mathbf{z})\|}\right)$, *as the angle between the l-embedding vectors* $\alpha^{(l)}(\mathbf{x})$ *and* $\alpha^{(l)}(\mathbf{z})$, *and the model gradient angle,* $\phi(\mathbf{x}, \mathbf{z}) \triangleq \arccos\left(\frac{\langle \nabla f(\mathbf{x}), \nabla f(\mathbf{x})(\mathbf{z}) \rangle}{\|\nabla f(\mathbf{x})(\mathbf{x})\|\|\nabla f(\mathbf{x})(\mathbf{z})\|}\right)$, *as the angle between the model gradient vectors* $\nabla f(\mathbf{x})$ *and* $\nabla f(\mathbf{z})$.

We also denote $\theta^{(0)}$ by $\theta_{in}$, as $\theta^{(0)}$ is just the angle between the original inputs.

In the rest of the paper, we specifically refer the NTK matrix, NTK condition number, $l$-embedding angle and model gradient angle for the ReLU neural network as $K$, $\kappa$, $\theta^{(l)}$ and $\phi$, respectively, and refer their linear neural network counterparts as $\bar{K}$, $\bar{\kappa}$, $\bar{\theta}^{(l)}$ and $\bar{\phi}$, respectively. We also denote the condition number of Gram matrix $G$ by $\kappa_0$.

## 3 LINEAR NEURAL NETWORK: THE BASELINE FOR COMPARISON

To distill the effect of the non-linear activation function, we need a activation-free case as the baseline for comparison. This baseline is the linear neural network $\bar{f}$, with the same width and depth as $f$.

**Theorem 3.1.** *Consider the linear neural network* $\bar{f}$ *as defined in Eq.(3). In the limit of infinite network width* $m \to \infty$ *and at network initialization* $\mathbf{w}_0$, *the following relations hold:*

- *for any input* $\mathbf{x} \in \mathbb{R}^d$: $\|\bar{\alpha}^{(l)}(\mathbf{x})\| = \|\mathbf{x}\|, \forall l \in [L]$; *and* $\|\nabla f(\mathbf{w}_0; \mathbf{x})\| = (L+1)\|\mathbf{x}\|$.

- *for any inputs* $\mathbf{x}, \mathbf{z} \in \mathbb{R}^d$: $\bar{\theta}^{(l)}(\mathbf{x}, \mathbf{z}) = \theta_{in}(\mathbf{x}, \mathbf{z}), \forall l \in [L]$; *and* $\bar{\phi}(\mathbf{x}, \mathbf{z}) = \theta_{in}(\mathbf{x}, \mathbf{z})$.

This theorem states that, without a non-linear activation function, both the feature embedding maps $\alpha^{(l)} : \mathbf{x} \mapsto \alpha^{(l)}(\mathbf{x})$ and the model gradient map $\nabla f : \mathbf{x} \mapsto \nabla f(\mathbf{x})$ fail to change the geometrical relationship between any data samples. For any input pairs, the embedding angles $\bar{\theta}^{(l)}$ and $\bar{\phi}$ remain the same as the input angle $\theta_{in}$. Therefore, it is not surprising that the NTK of a linear network is the same as the Gram matrix (up to a constant factor), as formally stated in the following corollary.

**Corollary 3.2** (NTK condition number of linear networks). *Consider a linear neural network* $\bar{f}$ *as defined in Eq.(3). In the limit of infinite network width* $m \to \infty$ *and at network initialization, the NTK matrix* $\bar{K} = (L+1)^2 G$. *Moreover,* $\bar{\kappa} = \kappa_0$.

This corollary tells that, for a linear neural network, regardless of its depth $L$, the NTK condition number $\bar{\kappa}$ is always equal to the condition number $\kappa_0$ of the Gram matrix $G$. Therefore, any non-zero deviations, $\delta\phi \triangleq \phi - \theta_{in}$ from the input angle $\theta_{in}$, and $\delta\kappa \triangleq \kappa - \kappa_0$ from the Gram condition number $\kappa_0$, observed for a non-linearly activated network $f$, should be attributed to the corresponding non-linear activation.

## 4  ReLU induces better data separation in model gradient space

In this section, we show that the ReLU non-linearity helps data separation in the model gradient space. Specifically, for two arbitrary inputs $\mathbf{x}$ and $\mathbf{z}$ with small $\theta_{in}(\mathbf{x}, \mathbf{z})$, we show that the model gradient angle $\phi(\mathbf{x}, \mathbf{z})$ is strictly larger than $\theta_{in}(\mathbf{x}, \mathbf{z})$, implying a better angle separation of the two data points in the model gradient space. Moreover, we show that the model gradient angle $\phi(\mathbf{x}, \mathbf{z})$ monotonically increases with the number of layers $L$, indicating that deeper network (more ReLU non-linearity) has better angle separation.

**Embedding vectors and embedding angles.**  We start with investigating the relations among the $l$-embedding vectors $\alpha^{(l)}$ and the embedding angles $\theta^{(l)}$.

**Lemma 4.1.** *Consider the ReLU network $f$ defined in Eq.(1) at its initialization, and define function $g : [0, \pi] \to [0, \pi]$ as $g(z) = \arccos\left(\frac{\pi - z}{\pi} \cos z + \frac{1}{\pi} \sin z\right)$. In the infinite network width limit $m \to \infty$, for all $l \in [L]$, the following relations hold:*

- *for any input $\mathbf{x} \in \mathbb{R}^d$, $\|\alpha^{(l)}(\mathbf{x})\| = \|\mathbf{x}\|$;*

- *for any two inputs $\mathbf{x}, \mathbf{z} \in \mathbb{R}^d$, $\theta^{(l)}(\mathbf{x}, \mathbf{z}) = g\left(\theta^{(l-1)}(\mathbf{x}, \mathbf{z})\right)$. Let $g^l(\cdot)$ be the $l$-fold composition of $g(\cdot)$, then*

$$\theta^{(l)}(\mathbf{x}, \mathbf{z}) = g^l\left(\theta_{in}(\mathbf{x}, \mathbf{z})\right). \tag{7}$$

The lemma states that, during forward propagation, the $l$-embedding vectors for each input keeps unchanged in magnitude, and the embedding angles $\theta^{(l)}$ between any two inputs are governed by the closed form function $g$. Please see Appendix A for the plot of the function and detailed discussion about its properties. As a highlight, $g$ has the following property: $g$ is approximately the identity function $g(z) \approx z$ for small $z$, i.e., $z \ll 1$. This property directly implies the following theorem.

**Theorem 4.2.** *Given any inputs $\mathbf{x}, \mathbf{z}$ such that $\theta_{in}(\mathbf{x}, \mathbf{z}) = o(1/L)$, for each $l \in [L]$, the $l$-embedding angle $\theta^{(l)}(\mathbf{x}, \mathbf{z})$ can be expressed as*

$$\theta^{(l)}(\mathbf{x}, \mathbf{z}) = \theta_{in}(\mathbf{x}, \mathbf{z}) - \frac{l}{3\pi}(\theta_{in}(\mathbf{x}, \mathbf{z}))^2 + o\left((\theta_{in}(\mathbf{x}, \mathbf{z}))^2\right).$$

We see that, at the small angle regime $\theta_{in} = o(1/L)$, the embedding angles $\theta^{(l)}$ at any layer $l$ is the same as the input angle $\theta_{in}$ at the lowest order. In addition, the higher order corrections are always negative making $\theta^{(l)} < \theta_{in}$. We also note that the correction term $\Delta\theta^{(l)} \triangleq \theta^{(l)} - \theta_{in}$ is linearly dependent on layer $l$ at its lowest order.

**Model gradient angle.**  Now, we investigate the model gradient angle $\phi$ and its relation with the embedding angles $\theta^{(l)}$ and input angle $\theta_{in}$, for the ReLU network.

**Lemma 4.3.** *Consider the ReLU network defined in Eq.(1) with $L$ hidden layers and infinite network width $m$. Given two arbitrary inputs $\mathbf{x}$ and $\mathbf{z}$, the angle $\phi(\mathbf{x}, \mathbf{z})$ between the model gradients $\nabla f(\mathbf{x})$ and $\nabla f(\mathbf{z})$ satisfies*

$$\cos\phi(\mathbf{x}, \mathbf{z}) = \frac{1}{L+1} \sum_{l=0}^{L} \left[\cos\theta^{(l)}(\mathbf{x}, \mathbf{z}) \prod_{l'=l}^{L-1} (1 - \theta^{(l')}(\mathbf{x}, \mathbf{z})/\pi)\right]. \tag{8}$$

*Moreover, $\|\nabla f(\mathbf{x})\| = (L+1)\|\mathbf{x}\|$, for any $\mathbf{x}$.*

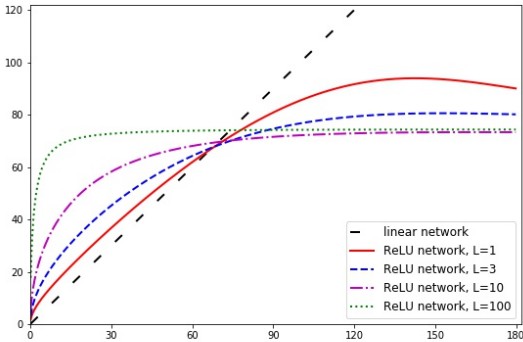

Figure 1: **Model gradient angles $\phi$ vs. input angle $\theta_{in}$ (according to Lemma 4.3).** Linear neural networks, of any depth $L$, always have $\bar{\phi} = \theta_{in}$, as the black dash line showed. ReLU neural networks with various depths have better data separation $\phi > \theta_{in}$ for similar data (i.e., small $\theta_{in}$). Moreover, deeper ReLU networks have better separation than shallow ones for similar data. All neural networks are infinitely wide.

**Better data separation with ReLU.** Comparing with Theorem 3.1 for linear neural networks, we see that the non-linear ReLU activation only affects the relative direction, but not the the magnitude, of the model gradient. Combining Lemmas 4.3 and 4.1, we get the relation between $\phi$ and the input angle $\theta_{in}$. Figure 1 plots $\phi$ as a function of $\theta_{in}$ for different network depth $L$.

The **key observation** is that: for relatively small input angles (say, $\theta_{in} < 60°$), the model gradient angle $\phi$ is always greater than the input angle $\theta_{in}$. This suggests that, after the mapping $\nabla f : \mathbf{x} \mapsto \nabla f(\mathbf{x})$ from the input space to model gradient space, data inputs becomes more (directionally) separated, if they are similar in the input space (i.e., with small $\theta_{in}$). Comparing to the linear neural network case, where $\bar{\phi}(\mathbf{x}, \mathbf{z}) = \theta_{in}(\mathbf{x}, \mathbf{z})$ as in Theorem 3.1, we see that the ReLU non-linearity results in a better angle separation $\phi(\mathbf{x}, \mathbf{z}) > \bar{\phi}(\mathbf{x}, \mathbf{z})$ for similar data.

Another important observation is that: deeper ReLU networks lead to larger model gradient angles, when $\theta_{in} < 60°$. This indicates that deeper ReLU networks, which has more layers of ReLU non-linear activation, makes the model gradient more separated between inputs. Note that, in the linear network case, the depth does not affect the gradient angle $\bar{\phi}$.

We theoretically confirm these two observations in the regime of small input angle $\theta_{in} = o(1/L)$, by the following theorem.

**Theorem 4.4** (Better separation with ReLU). *Consider two network inputs $\mathbf{x}, \mathbf{z} \in \mathbb{R}^d$, with small input angle $\theta_{in}(\mathbf{x}, \mathbf{z}) = o(1/L)$, and the ReLU network defined in Eq.(1) with $L$ hidden layers and infinite network width $m$. At the network initialization, the angle $\phi(\mathbf{x}, \mathbf{z})$ between the model gradients $\nabla f(\mathbf{x})$ and $\nabla f(\mathbf{z})$ satisfies*

$$\cos \phi(\mathbf{x}, \mathbf{z}) = \left(1 - \frac{L}{2\pi}\theta_{in} + o(\theta_{in})\right) \cos \theta_{in}. \tag{9}$$

Noticing the negative sign within the factor $\left(1 - \frac{L}{2\pi}\theta_{in} + o(\theta_{in})\right)$, we know that the factor is less than 1 and we obtain that: $\phi(\mathbf{x}, \mathbf{z}) > \theta_{in}(\mathbf{x}, \mathbf{z}) = \bar{\phi}(\mathbf{x}, \mathbf{z})$. Noticing the depth $L$ dependence of this factor, we also get that: the deeper the ReLU network (i.e., larger $L$) is, the larger $\phi$ is, in the regime $\theta_{in} = o(1/L)$.

**Remark 4.5** (Separation in distance). *Indeed, the better angle separation discussed above implies a better separation in Euclidean distance as well. This can be easily seen by recalling from Lemma 4.3 that the model gradient mapping $\nabla f$ preserves the norm (up to a universal factor $L + 1$).*

We also point out that, Figure 1 indicates that for large input angles (say $\theta_{in} > 60°$) the model gradient angle $\phi$ is always large (greater than $60°$). Hence, non-similar data never become similar in the model gradient feature space.

## 5 ReLU INDUCES SMALLER NTK CONDITION NUMBER OF NTK

In this section, we show both theoretically and experimentally that, ReLU induces a decrease in the NTK condition number $\kappa$. Moreover, a ReLU network with larger depth $L$, which means more non-linear activations in operation, the NTK condition number $\kappa$ is generically smaller.

### 5.1 THEORETICAL ANALYSIS

**Connection between condition number and model gradient angle.** The smallest eigen-value and condition number of NTK are closely related to the smallest model gradient angle $\min_{i,j\in[n]} \phi(\mathbf{x}_i, \mathbf{x}_j)$, through the gradient feature matrix $F$. Think about the case if $\phi(\mathbf{x}_i, \mathbf{x}_j) = 0$ (i.e., $\nabla f(\mathbf{x}_i)$ is parallel to $\nabla f(\mathbf{x}_j)$) for some $i, j \in [n]$, then $F$, hence NTK $K$, is not full rank and the smallest eigenvalue $\lambda_{min}(K)$ is zero, leading to an infinite condition number $\kappa$. Similarly, if $\min_{i,j\in[n]} \phi(\mathbf{x}_i, \mathbf{x}_j)$ is small, the smallest eigenvalue $\lambda_{min}(K)$ is also small, and condition number $\kappa$ is large, as stated in the following proposition (see proof in Appendix B).

**Proposition 5.1.** *Consider a $n \times n$ positive definite matrix $A = BB^T$, where matrix $B \in \mathbb{R}^{n \times d}$, with $d > n$, is of full row rank. Suppose that there exist $i, j \in [n]$ such that the angle $\phi$ between vectors $B_{i\cdot}$ and $B_{j\cdot}$ is small, i.e., $\phi \ll 1$, and that there exist constant $C > c > 0$ such that $c \leq \|B_{k\cdot}\| \leq C$ for all $k \in [n]$. Then, the smallest eigenvalue $\lambda_{min}(A) = O(\phi^2)$, and the condition number $\kappa = \Omega(1/\phi^2)$.*

Hence, a good data angle separation in the model gradient features, i.e., $\min_{i,j\in[n]} \phi(\mathbf{x}_i, \mathbf{x}_j)$ not too small, is a necessary condition such that the condition number $\kappa$ is not too large.

**Smaller NTK condition number.** Theoretically, we consider the infinite width limit. First, we consider the special case where the dataset is of size 2.

**Theorem 5.2.** *Consider a $L$-layer ReLU neural network $f$ as defined in Eq.(1) in the infinite width limit $m \to \infty$ and at initialization. Consider the dataset $\mathcal{D} = \{(\mathbf{x}_1, y_1), (\mathbf{x}_2, y_2)\}$ with the input angle $\theta_{in}$ between $\mathbf{x}_1$ and $\mathbf{x}_2$ small, $\theta_{in} = o(1/L)$. Then, the NTK condition number $\kappa < \kappa_0$. Moreover, for two ReLU neural networks $f_1$ of depth $L_1$ and $f_2$ of depth $L_2$ with $L_1 > L_2$, we have $\kappa_{f_1} < \kappa_{f_2}$.*

The non-linear ReLU makes the samples more separated when mapped from the input data space to the model gradient feature space. This theorem confirms the expectation that the NTK condition number $\kappa$ should be decreased, as a consequence of the existence of the ReLU non-linearity. This theorem also shows that the depth of ReLU network enhances this better NTK conditioning.

Indeed, the above dataset assumption is not a necessary requirement and can be alleviated. Now, let's look at the NTK of a shallow ReLU neural network

$$f(W; \mathbf{x}) = \frac{\sqrt{2}}{\sqrt{m}} \mathbf{v}^T \sigma(W\mathbf{x}). \tag{10}$$

Here, we fix $\mathbf{v}$ to random initialization and let $W$ trainable. We only require that the dataset is not degenerated, i.e., $\mathbf{x}_i \nparallel \mathbf{x}_j$ for all $i, j$.

**Theorem 5.3.** *Consider the ReLU network in Eq.(10) in the limit $m \to \infty$ and at initialization. The smallest eigenvalue $\lambda_{min}(K)$ of its NTK is larger than that of the Gram matrix: $\lambda_{min}(K) > \lambda_{min}(G)$. Moreover, the NTK condition number is less than that of the Gram matrix: $\kappa < \kappa_0$.*

The high-level intuition is that, when there are ReLU implemented, the model gradient map $\nabla f : \mathbf{x} \mapsto \nabla f(\mathbf{x})$ increases the directional diversity of vectors $\nabla f(\mathbf{x})$, thanks for the high dimension of model gradient space and the different activation patterns of the hidden layer for different samples $\mathbf{x}$. Hence, it is expected that the feature matrix $F$, as well as the NTK matrix $K$, is better conditioned.

### 5.2 EXPERIMENTAL EVIDENCE

In this subsection, we experimentally show that the phenomena of better data separation and better NTK conditioning widely happen in practice.

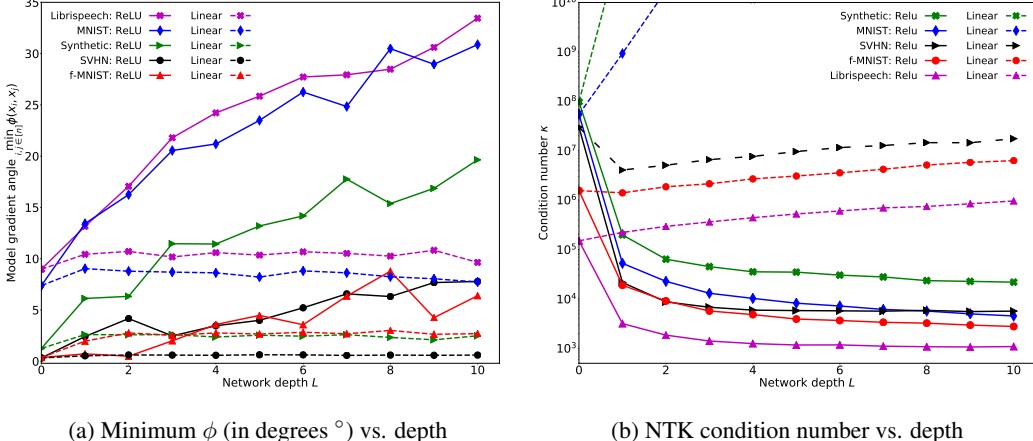

(a) Minimum $\phi$ (in degrees $^\circ$) vs. depth       (b) NTK condition number vs. depth

Figure 2: **Better separation (left) and better NTK conditioning (right) of ReLU network.** Solid lines are for ReLU networks, and dash lines are for linear networks. **Left:** ReLU network works better in separating similar data, while linear network remains similar to a linear model. **Right:** ReLU network has better conditioning of NTK than linear network and linear model. Note that $L = 0$ (without any hidden layer) corresponds to the case of a linear model, and the NTK in this case is the Gram matrix.

**Dataset.** We use the following datasets: synthetic dataset, MNIST (LeCun et al., 1998), FashionM-NIST (f-MNIST) (Xiao et al., 2017), SVHN (Netzer et al., 2011) and Librispeech (Panayotov et al., 2015). The synthetic data consists of 2000 samples which are randomly drawn from a 5-dimensional Gaussian distribution with zero-mean and unit variance. The MNIST, f-MNIST and SVHN datasets are image datasets where each input is an image. The Librispeech is a speech dataset including 100 hours of clean speeches. In the experiments, we use a subset of Librispeech with $50,000$ samples, and each input is a 768-dimensional vector representing a frame of speech audio and we follow (Hui & Belkin, 2020) for the feature extraction.

**Models.** For each of the datasets, we use a ReLU activated fully-connected neural network architecture to process. The ReLU network has $L$ hidden layers, and has $512$ neurons in each of its hidden layers. The ReLU network uses the NTK parameterization and initialization strategy (see (Jacot et al., 2018)). For each dataset, we vary the network depth $L$ from 0 to 10. Note that $L = 0$ corresponding to the linear model case. In addition, for comparison, we use a linear neural network, which has the same architecture with the ReLU network except the absence of activation function.

**Results.** For each of the experimental setting, we evaluate both the smallest pairwise model gradient angle $\min_{i,j\in[n]} \phi(\mathbf{x}_i, \mathbf{x}_j)$ and the NTK condition number $\kappa$, at the network initialization. We take 5 independent runs over 5 random initialization seeds, and report the average. The results are shown in Figure 2. As one can easily see from the plots, a ReLU network (depth $L = 1, 2, \cdots, 10$) always have a better separation of data (i.e., larger smallest pairwise model gradient angle), and a better NTK conditioning (i.e., smaller NTK condition number), than its corresponding linear network (compare the solid line and dash line of the same color). Furthermore, the monotonically decreasing NTK condition number shows that a deeper ReLU network have a better conditioning of NTK.

## 6 OPTIMIZATION ACCELERATION

Recently studies showed strong connections between the NTK condition number and the theoretical convergence rate of gradient descent algorithms on wide neural networks (Du et al., 2018; 2019a; Soltanolkotabi et al., 2018; Allen-Zhu et al., 2019; Zou et al., 2020; Oymak & Soltanolkotabi, 2020; Liu et al., 2022). In Du et al. (2018; 2019a), the authors derived the worst-case convergence rates explicitly in terms of the smallest eigenvalue of NTK $\lambda_{min}(K)$, $L(\mathbf{w}_t) \leq (1 - \eta\lambda_{min}(K)/2)^t L(\mathbf{w}_0)$, where $L$ is the square loss function and $t$ is the time stamp of the algorithm. Later on, in Liu et al.

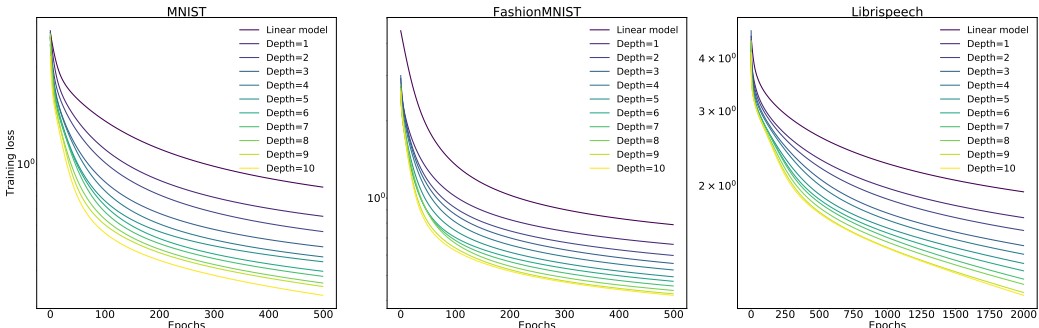

Figure 3: **Training curve of ReLU networks with different depths.** On each of these datasets, we see that deeper ReLU network always converges faster than shallower ones.

(2022), the NTK condition number is explicitly involved in the convergence rate:

$$L(\mathbf{w}_t) \leq (1 - \kappa^{-1})^t L(\mathbf{w}_0). \tag{11}$$

Although $\kappa$ is evaluated on the whole optimization path, all these theories used the fact that NTK is almost constant for wide neural networks and an evaluation at the initialization $\mathbf{w}_0$ is enough.

As a smaller NTK condition number (or larger smallest eigenvalue of NTK) implies a faster worst-case convergence rate, our findings suggest that: (a), the ReLU activation function helps improving the worst-case convergence rate of gradient descent, and (b), deeper wide ReLU networks have faster convergence rate than shallower ones.

We experimentally verify this implication. Specifically, we train the ReLU networks, with depth $L$ ranging from 1 to 10, for the datasets MNIST, f-MNIST and Librispeech. For all the training tasks, we use cross entropy loss as the objective function and use mini-batch stochastic gradient descent (SGD) of batch size 500 to optimize. For each task, we find its optimal learning rate by grid search. On MNIST and f-MNIST, we train 500 epochs, and on Librispeech, we training 2000 epochs.

The curves of training loss against epochs are shown in Figure 3. We observe that, for all these datasets, a deeper ReLU network always converges faster than shallower ones. This is consistent with the theoretical prediction that the deeper ReLU network, which has smaller NTK condition number, has faster theoretical convergence rate.

## 7    CONCLUSION AND DISCUSSIONS

In this work, we showed the beneficial effects of ReLU non-linear activation on the data separation in feature space and on the NTK conditioning. We also showed that more sequential ReLU activation operations, i.e., larger network depth, magnifies these effects. As the NTK conditioning is closely related to theoretical convergence rate of gradient descent, our findings also suggest a positive role of the ReLU activation function in optimization theories.

**Finite network width.**   Our theoretical analysis is based the setting of infinite network width, under which setting the analysis becomes simplified. As our experiments showed, the same results also hold on the finite width case across various tasks. For finite but large network width and at network initialization, the analysis only differs by a small zero-mean noisy term, wherever we took the limit $m \to \infty$. We believe, with a quantitative analysis that controls these small terms and the deviations $\delta\phi$ and $\delta\kappa$, our results still hold. We leave it as a future work.

**Infinite depth.**   In this work, we focused on the finite depth scenario which is the more interesting case from a practical point of view. Our small angle regime analysis (Theorem 4.2, 4.4 and 5.2) do not directly extend to the infinite depth case. But, as Lemma 4.3 and Figure 1 indicate, the $\phi(\theta_{in})$ function seems to converge to a step function when $L \to \infty$, which implies orthogonality between model gradient vectors, hence a NTK condition number being 1. This is consistent with the prior knowledge that NTK converges to 1 in the infinite depth limit (Radhakrishnan et al., 2023).

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
