## A  PROPERTIES OF FUNCTION $g$

Recall that the function $g : [0, \pi) \to [0, \pi)$ is defined as (see Lemma 4.1)

$$g(z) = \arccos\left(\frac{\pi - z}{\pi} \cos z + \frac{1}{\pi} \sin z\right), \tag{12}$$

Figure 4 shows the plot of this function. From the plot, we can easily find the following properties.

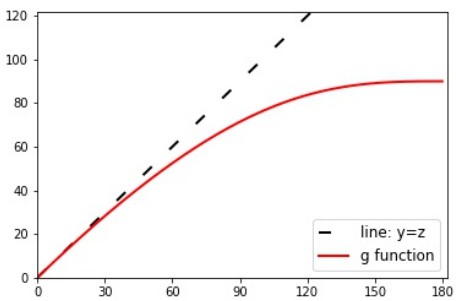

Figure 4: Curve of the function $g(\theta)$. As can be seen, $g(\theta)$ is monotonic, and is approximately the identity function $y = \theta$ in the small angle region ($\theta \ll 90°$).

**Proposition A.1** (Properties of $g$). *The function $g$ defined in Eq.(12) has the following properties:*

1. *$g$ is a monotonically increasing function;*

2. *$g(z) \leq z$, for all $z \in [0, \pi)$; and $g(z) = z$ if and only if $z = 0$;*

3. *for any $z \in [0, \pi)$, the sequence $\{g^l(z)\}_{l=1}^{\infty}$ is monotonically decreasing, and has the limit $\lim_{l \to \infty} g^l(z) = 0$.*

It is worth to note that the last property of $g$ function immediately implies the collapse of embedding vectors from different inputs in the infinite depth limit $L \to \infty$. This embedding collapse has been observed in prior works Poole et al. (2016); Schoenholz et al. (2016) (although by different type of analysis) and has been widely discussed in the literature of Edge of Chaos.

**Theorem A.2.** *Consider the same ReLU neural network as in Lemma 4.1. Given any two inputs $\mathbf{x}, \mathbf{z} \in \mathbb{R}^d$, the sequence of angles between their $l$-embedding vectors, $\{\theta^{(l)}(\mathbf{x}, \mathbf{z})\}_{l=1}^{L}$, is monotonically decreasing. Moreover, in the limit of infinite depth,*

$$\lim_{L \to \infty} \theta^{(L)}(\mathbf{x}, \mathbf{z}) = 0, \tag{13}$$

*and there exists a vector $\alpha$ such that, for any input $\mathbf{x}$, the last layer $L$-embedding*

$$\alpha^{(L)}(\mathbf{x}) = \|\mathbf{x}\|\alpha. \tag{14}$$

*Proof of Proposition A.1.* **Part 1.** First, we consider the auxiliary function $\tilde{g}(z) = \frac{\pi - z}{\pi} \cos z + \frac{1}{\pi} \sin z$. We see that

$$\frac{d\tilde{g}(z)}{dz} = -\left(1 - \frac{z}{\pi}\right) \sin z \leq 0, \quad \forall z \in [0, \pi).$$

Hence, $\tilde{g}(z)$ is monotonically decreasing on $[0, \pi)$. Combining with the monotonically decreasing nature of the $\arccos$ function, we get that $g$ is monotonically increasing.

**Part 2.** It suffices to prove that $\cos z \leq \tilde{g}(z)$ and that the equality holds only at $z = 0$. For $z = 0$, it is easy to check that $\cos z = \tilde{g}(z)$, as both $z$ and $\sin z$ are zero. For $z \in (0, \pi/2)$, noting that $\tan z - z > 0$, we have

$$\tilde{g}(z) = \frac{\pi - z}{\pi} \cos z + \frac{1}{\pi} \sin z = \cos z + \frac{1}{\pi}(-z + \tan z) \cos z > \cos z. \tag{15}$$

For $z = \pi/2$, we have $\cos \pi/2 = 0 < 1/\pi = \tilde{g}(\pi/2)$. For $z \in (\pi/2, \pi)$, we have the same relation as in Eq.(15). The only differences are that, in this case, $\cos z < 0$ and $\tan z - z < 0$. Therefore, we still get $\tilde{g}(z) > \cos z$ for $z \in (\pi/2, \pi)$.

**Part 3.** From part 2, we see that $g(z) < z$ for all $z \in (0, \pi)$. Hence, for any $l$, $g^{l+1}(z) < g^l(z)$. Moreover, since $z = 0$ is the only fixed point such that $g(z) = z$, in the limit $l \to \infty$, $g^l(z) \to 0$. $\quad\square$

## B  PROOF OF PROPOSITION 5.1

*Proof.* Consider the matrix $B$ and the $n$ vectors $\mathbf{b}_k \triangleq B_{k\cdot}$, $k \in [n]$. The smallest singular value square of matrix $B$ is defined as

$$\sigma_{min}^2(B) = \min_{\mathbf{v} \neq 0} \frac{\mathbf{v}^T B B^T \mathbf{v}}{\mathbf{v}^T \mathbf{v}} = \min_{\mathbf{v} \neq 0} \frac{\|\sum_k v_k \mathbf{b}_k\|^2}{\|\mathbf{v}\|^2}.$$

Since the angle $\phi$ between $\mathbf{b}_i = B_{i\cdot}$ and $\mathbf{b}_j = B_{j\cdot}$ is small, let $\mathbf{v}'$ be the vector such that $v_i' = \|\mathbf{b}_j\|$, $v_j' = -\|\mathbf{b}_i\|$ and $v_k' = 0$ for all $k \neq i, j$. Then

$$
\begin{aligned}
\sigma_{min}^2(B) &\leq \frac{\|\sum_k v_k' \mathbf{b}_k\|^2}{\|\mathbf{v}'\|^2} = \left\| \frac{\|\mathbf{b}_j\|}{\sqrt{\|\mathbf{b}_i\|^2 + \|\mathbf{b}_j\|^2}} \mathbf{b}_i - \frac{\|\mathbf{b}_i\|}{\sqrt{\|\mathbf{b}_i\|^2 + \|\mathbf{b}_j\|^2}} \mathbf{b}_j \right\|^2 \\
&= \frac{2\|\mathbf{b}_i\|^2 \|\mathbf{b}_j\|^2}{\|\mathbf{b}_i\|^2 + \|\mathbf{b}_j\|^2} (1 - \cos\phi) \\
&= \frac{\|\mathbf{b}_i\|^2 \|\mathbf{b}_j\|^2}{\|\mathbf{b}_i\|^2 + \|\mathbf{b}_j\|^2} \phi^2 + O(\phi^4).
\end{aligned}
$$

Since $A = BB^T$, the smallest eigenvalue $\lambda_{min}(A)$ of $A$ is the same as $\sigma_{min}^2(B)$.

On the other hand, the largest eigenvalue $\lambda_{max}(A)$ of matrix $A$ is lower bounded by $\text{tr}(A)/n$. Note that the diagonal entries $A_{kk} = \|\mathbf{b}_k\|$. Hence, $c \leq \lambda_{max}(A) \leq C$. Therefore, the condition number $\kappa = \lambda_{max}(A)/\lambda_{min}(A) = \Omega(1/\phi^2)$. $\quad\square$

## C  PROOFS OF THEOREMS FOR LINEAR NEURAL NETWORK

### C.1  PROOF OF THEOREM 3.1

*Proof.* First of all, we provide a useful lemma.

**Lemma C.1.** *Consider a matrix $A \in \mathbb{R}^{m \times d}$, with each entry of $A$ is i.i.d. drawn from $\mathcal{N}(0, 1)$. In the limit of $m \to \infty$,*

$$\frac{1}{m} A^T A \to I_{d \times d}, \quad \text{in probability.} \tag{16}$$

We first consider the embedding vectors $\bar{\alpha}^{(l)}$ and the embedding angles $\bar{\theta}^{(l)}$. By definition in Eq.(3), we have, for all $l \in [L]$ and input $\mathbf{x} \in \mathbb{R}^d$,

$$\bar{\alpha}^{(l)}(\mathbf{x}) = \frac{1}{m^{l/2}} W^{(l)} W^{(l-1)} \cdots W^{(1)} \mathbf{x}. \tag{17}$$

Note that at the network initialization entries of $W^{(l)}$ are i.i.d. and follows $\mathcal{N}(0, 1)$. Hence, the inner product

$$\langle \bar{\alpha}^{(l)}(\mathbf{x}), \bar{\alpha}^{(l)}(\mathbf{z}) \rangle = \frac{1}{m^l} \mathbf{x}^T W^{(1)T} \cdots W^{(l-1)T} W^{(l)T} W^{(l)} W^{(l-1)} \cdots W^{(1)} \mathbf{z} \overset{(a)}{=} \mathbf{x}^T \mathbf{z},$$

where in step (a) we recursively applied Lemma C.1 $l$ times. Putting $\mathbf{z} = \mathbf{x}$, we get $\|\bar{\alpha}^{(l)}(\mathbf{x})\| = \|\mathbf{x}\|$, for all $l \in [L]$. By the definition of embedding angles, it is easy to check that $\bar{\theta}^{(l)}(\mathbf{x}, \mathbf{z}) = \theta_{in}(\mathbf{x}, \mathbf{z})$, for all $l \in [L]$.

Now, we consider the model gradient $\nabla \bar{f}$ and the model gradient angle $\bar{\phi}$. As we consider the model gradient only at network initialization, we don't explicitly write out the dependence on $\mathbf{w}_0$, and we write $\nabla \bar{f}(\mathbf{w}_0, \mathbf{x})$ simply as $\nabla \bar{f}(\mathbf{x})$. The model gradient $\nabla \bar{f}$ can be decomposed as

$$\nabla \bar{f}(\mathbf{x}) = (\nabla_1 \bar{f}(\mathbf{x}), \nabla_2 \bar{f}(\mathbf{x}), \cdots, \nabla_{L+1} \bar{f}(\mathbf{x})), \;\; with \; \nabla_l \bar{f}(\mathbf{x}) = \frac{\partial \bar{f}(\mathbf{x})}{\partial W^{(l)}}, \forall l \in [L+1]. \quad (18)$$

Hence, the inner product

$$\langle \nabla \bar{f}(\mathbf{x}), \nabla \bar{f}(\mathbf{z}) \rangle = \sum_{l=1}^{L+1} \langle \nabla_l \bar{f}(\mathbf{x}), \nabla_l \bar{f}(\mathbf{z}) \rangle,$$

and for all $l \in [l+1]$,

$$\langle \nabla_l \bar{f}(\mathbf{x}), \nabla_l \bar{f}(\mathbf{z}) \rangle = \langle \bar{\alpha}^{(l-1)}(\mathbf{x}), \bar{\alpha}^{(l-1)}(\mathbf{z}) \rangle \cdot \langle \prod_{l'=l+1}^{L+1} \frac{1}{\sqrt{m}} W^{(l')T}, \prod_{l'=l+1}^{L+1} \frac{1}{\sqrt{m}} W^{(l')T} \rangle \overset{(b)}{=} \mathbf{x}^T \mathbf{z}.$$

Here in step (b), we again applied Lemma C.1. Therefore,

$$\langle \nabla \bar{f}(\mathbf{x}), \nabla \bar{f}(\mathbf{z}) \rangle = (L+1) \mathbf{x}^T \mathbf{z}. \quad (19)$$

Putting $\mathbf{z} = \mathbf{x}$, we get $\|\nabla f(\mathbf{x})\| = (L+1)\|\mathbf{x}\|$. By the definition of model gradient angle, it is easy to check that $\bar{\phi}(\mathbf{x}, \mathbf{z}) = \theta_{in}(\mathbf{x}, \mathbf{z})$. □

# D   PROOFS OF THEOREMS FOR RELU NETWORK

## D.1   PRELIMINARY RESULTS

Before the proofs, we introduce some useful notations and lemmas.

Given a vector $\mathbf{v} \in \mathbb{R}^p$, we define the following diagonal indicator matrix:

$$\mathbb{I}_{\{\mathbf{v} \geq 0\}} = \mathsf{diag}\left(\mathbb{I}_{\{v_1 \geq 0\}}, \mathbb{I}_{\{v_2 \geq 0\}}, \cdots, \mathbb{I}_{\{v_p \geq 0\}}\right), \quad (20)$$

with

$$\mathbb{I}_{\{v_i \geq 0\}} = \begin{cases} 1 & v_i \geq 0, \\ 0 & v_i < 0. \end{cases}$$

**Lemma D.1.** *Consider two vectors $\mathbf{v}_1, \mathbf{v}_2 \in \mathbb{R}^p$ and a $p$-dimensional random vector $\mathbf{w} \sim \mathcal{N}(0, I_{p \times p})$. Denote $\theta$ as the angle between $\mathbf{v}_1$ and $\mathbf{v}_2$, i.e., $\cos\theta = \frac{\langle \mathbf{v}_1, \mathbf{v}_2 \rangle}{\|\mathbf{v}_1\|\|\mathbf{v}_2\|}$. Then, the probability*

$$\mathbb{P}[(\mathbf{w}^T \mathbf{v}_1 \geq 0) \wedge (\mathbf{w}^T \mathbf{v}_2 \geq 0)] = \frac{1}{2} - \frac{\theta}{2\pi}. \quad (21)$$

**Lemma D.2.** *Consider two arbitrary vectors $\mathbf{v}_1, \mathbf{v}_2 \in \mathbb{R}^p$ and a random matrix $W \in \mathbb{R}^{q \times p}$ with entries $W_{ij}$ i.i.d. drawn from $\mathcal{N}(0,1)$. Denote $\theta$ as the angle between $\mathbf{v}_1$ and $\mathbf{v}_2$, and define $\mathbf{u}_1 = \frac{\sqrt{2}}{\sqrt{q}} \sigma(W\mathbf{v}_1)$ and $\mathbf{u}_2 = \frac{\sqrt{2}}{\sqrt{q}} \sigma(W\mathbf{v}_2)$. Then, in the limit of $q \to \infty$,*

$$\langle \mathbf{u}_1, \mathbf{u}_2 \rangle = \frac{1}{\pi}\left((\pi - \theta)\cos\theta + \sin\theta\right)\|\mathbf{v}_1\|\|\mathbf{v}_2\|. \quad (22)$$

**Lemma D.3.** *Consider two arbitrary vectors $\mathbf{v}_1, \mathbf{v}_2 \in \mathbb{R}^p$ and two random matrices $U \in \mathbb{R}^{s \times q}$ and $W \in \mathbb{R}^{q \times p}$, where all entries $U_{ij}$, $i \in [s]$ and $j \in [q]$, and $W_{kl}$, $k \in [q]$ and $l \in [p]$, are i.i.d. drawn from $\mathcal{N}(0,1)$. Denote $\theta$ as the angle between $\mathbf{v}_1$ and $\mathbf{v}_2$, and define matrices $A_1 = \frac{\sqrt{2}}{\sqrt{q}} U \mathbb{I}_{\{W\mathbf{v}_1 \geq 0\}}$ and $A_2 = \frac{\sqrt{2}}{\sqrt{q}} U \mathbb{I}_{\{W\mathbf{v}_2 \geq 0\}}$. Then, in the limit of $q \to \infty$, the matrix*

$$A_1 A_2^T = \frac{\pi - \theta}{\pi} I_{s \times s}. \quad (23)$$

## D.2 PROOF OF LEMMA 4.1

*Proof.* Consider an arbitrary layer $l \in [L]$ of the ReLU neural network $f$ at initialization. Given two arbitrary network inputs $\mathbf{x}, \mathbf{z} \in \mathbb{R}^d$, the inputs to the $l$-th layer are $\alpha^{(l-1)}(\mathbf{x}))$ and $\alpha^{(l-1)}(\mathbf{z}))$, respectively.

By definition, we have

$$\alpha^{(l)}(\mathbf{x}) = \sqrt{\frac{2}{m}}\sigma\left(W^{(l)}\alpha^{(l-1)}(\mathbf{x})\right), \quad \alpha^{(l)}(\mathbf{z}) = \sqrt{\frac{2}{m}}\sigma\left(W^{(l)}\alpha^{(l-1)}(\mathbf{z})\right), \tag{24}$$

with entries of $W^{(l)}$ being i.i.d. drawn from $\mathcal{N}(0, 1)$. Recall that, by definition, the angle between $\alpha^{(l-1)}(\mathbf{x}))$ and $\alpha^{(l-1)}(\mathbf{z}))$ is $\theta^{(l-1)}(\mathbf{x}, \mathbf{z})$. Applying Lemma D.2, we immediately have the inner product

$$\langle\alpha^{(l)}(\mathbf{z}), \alpha^{(l)}(\mathbf{x})\rangle = \frac{1}{\pi}\left((\pi - \theta^{(l-1)}(\mathbf{x}, \mathbf{z}))\cos\theta^{(l-1)}(\mathbf{x}, \mathbf{z}) + \sin\theta^{(l-1)}(\mathbf{x}, \mathbf{z})\right)$$
$$\times \|\alpha^{(l-1)}(\mathbf{x})\|\|\alpha^{(l-1)}(\mathbf{z})\|. \tag{25}$$

In the special case of $\mathbf{x} = \mathbf{z}$, we have $\theta^{(l-1)}(\mathbf{x}, \mathbf{z}) = 0$, and obtain from the above equation that

$$\|\alpha^{(l)}(\mathbf{x})\|^2 = \|\alpha^{(l-1)}(\mathbf{x})\|^2. \tag{26}$$

Apply Eq.(26) back to Eq.(25), we also get

$$\cos\theta^{(l)}(\mathbf{x}, \mathbf{z}) = \frac{\langle\alpha^{(l)}(\mathbf{z}), \alpha^{(l)}(\mathbf{x})\rangle}{\|\alpha^{(l)}(\mathbf{x})\|\|\alpha^{(l)}(\mathbf{z})\|} = \frac{1}{\pi}\left((\pi - \theta^{(l-1)}(\mathbf{x}, \mathbf{z}))\cos\theta^{(l-1)}(\mathbf{x}, \mathbf{z}) + \sin\theta^{(l-1)}(\mathbf{x}, \mathbf{z})\right) \tag{27}$$

That is $\theta^{(l)}(\mathbf{x}, \mathbf{z}) = g(\theta^{(l-1)}(\mathbf{x}, \mathbf{z}))$. Recursively apply this relation, we obtain the desired result. $\quad\square$

## D.3 PROOF OF THEOREM 4.2

*Proof.* By Lemma 4.1, we have that

$$\cos\theta^{(l)}(\mathbf{x}, \mathbf{z}) = \left(1 - \frac{\theta^{(l-1)}(\mathbf{x}, \mathbf{z})}{\pi}\right)\cos\theta^{(l-1)}(\mathbf{x}, \mathbf{z}) + \frac{1}{\pi}\sin\theta^{(l-1)}(\mathbf{x}, \mathbf{z})$$

$$= \cos\theta^{(l-1)}(\mathbf{x}, \mathbf{z})\left(1 + \frac{1}{\pi}\left(\tan\theta^{(l-1)}(\mathbf{x}, \mathbf{z}) - \theta^{(l-1)}(\mathbf{x}, \mathbf{z})\right)\right)$$

$$= \cos\theta^{(l-1)}(\mathbf{x}, \mathbf{z})\left(1 + \frac{1}{3\pi}(\theta^{(l-1)}(\mathbf{x}, \mathbf{z}))^3 + o\left((\theta^{(l-1)}(\mathbf{x}, \mathbf{z}))^3\right)\right).$$

Noting that the Taylor expansion of the $\cos$ function at zero is $\cos z = 1 - \frac{1}{2}z^2 + o(z^3)$, one can easily check that, for all $l \in [L]$,

$$\theta^{(l)}(\mathbf{x}, \mathbf{z}) = \theta^{(l-1)}(\mathbf{x}, \mathbf{z}) - \frac{1}{3\pi}(\theta^{(l-1)}(\mathbf{x}, \mathbf{z}))^2 + o\left((\theta^{(l-1)}(\mathbf{x}, \mathbf{z}))^2\right). \tag{28}$$

Note that $\theta^{(l)}(\mathbf{x}, \mathbf{z}) \leq \theta^{(l-1)}(\mathbf{x}, \mathbf{z}) = o(1/L)$. Iteratively apply the above equation, one gets, for all $l \in [L]$, if $\theta^{(0)}(\mathbf{x}, \mathbf{z}) = o(1/L)$,

$$\theta^{(l)}(\mathbf{x}, \mathbf{z}) = \theta^{(0)}(\mathbf{x}, \mathbf{z}) - \frac{l}{3\pi}(\theta^{(0)}(\mathbf{x}, \mathbf{z}))^2 + o\left((\theta^{(0)}(\mathbf{x}, \mathbf{z}))^2\right). \tag{29}$$

$$\square$$

## D.4 PROOF OF LEMMA 4.3

*Proof.* The model gradient $\nabla f(\mathbf{x})$ is composed of the components $\nabla_l f(\mathbf{x}) \triangleq \frac{\partial f}{\partial W^l}$, for $l \in [L+1]$. Each such component has the following expression: for $l \in [L+1]$

$$\nabla_l f(\mathbf{x}) = \alpha^{(l-1)}(\mathbf{x})\delta^{(l)}(\mathbf{x}), \tag{30}$$

where

$$\delta^{(l)}(\mathbf{x}) = \left(\frac{2}{m}\right)^{\frac{L-l+1}{2}} W^{(L+1)}\mathbb{I}_{\{\tilde{\alpha}^{(L)}(\mathbf{x})\geq 0\}}W^{(L)}\mathbb{I}_{\{\tilde{\alpha}^{(L-1)}(\mathbf{x})\geq 0\}}\cdots W^{(l+1)}\mathbb{I}_{\{\tilde{\alpha}^{(l)}(\mathbf{x})\geq 0\}}. \quad (31)$$

Note that in Eq.(30), $\nabla_l f(\mathbf{x})$ is an outer product of a column vector $\alpha^{(l-1)}(\mathbf{x}) \in \mathbb{R}^{m_{l-1}\times 1}$ ($m_{l-1} = d$ if $l = 1$, and $m_{l-1} = m$ otherwise) and a row vector $\delta^{(l)}(\mathbf{x}) \in \mathbb{R}^{1\times m_l}$ ($m_l = 1$ if $l = L+1$, and $m_l = m$ otherwise).

First, we consider the inner product $\langle \nabla_l f(\mathbf{z}), \nabla_l f(\mathbf{x})\rangle$, for $l \in [L+1]$.[1] By Eq.(30), we have

$$\langle \nabla_l f(\mathbf{z}), \nabla_l f(\mathbf{x})\rangle = \langle \delta^{(l)}(\mathbf{z}), \delta^{(l)}(\mathbf{x})\rangle \cdot \langle \alpha^{(l-1)}(\mathbf{z}), \alpha^{(l-1)}(\mathbf{x})\rangle. \quad (32)$$

For $\langle \alpha^{(l-1)}(\mathbf{z}), \alpha^{(l-1)}(\mathbf{x})\rangle$, applying Lemma 4.1, we have

$$\langle \alpha^{(l-1)}(\mathbf{z}), \alpha^{(l-1)}(\mathbf{x})\rangle = \|\mathbf{x}\|\|\mathbf{z}\|\cos\theta^{(l-1)}(\mathbf{x}, \mathbf{z}). \quad (33)$$

For $\langle \delta^{(l)}(\mathbf{z}), \delta^{(l)}(\mathbf{x})\rangle$, by definition Eq.(31), we have

$$\langle \delta^{(l)}(\mathbf{z}), \delta^{(l)}(\mathbf{x})\rangle = \left(\frac{2}{m}\right)^{L-l+1}$$
$$\times W^{(L+1)}\mathbb{I}_{\{\tilde{\alpha}^{(L)}(\mathbf{x})\geq 0\}}\cdots \underbrace{W^{(l+1)}\mathbb{I}_{\{\tilde{\alpha}^{(l)}(\mathbf{x})\geq 0, \tilde{\alpha}^{(l)}(\mathbf{z})\geq 0\}}W^{(l+1)T}}_{A}\cdots \mathbb{I}_{\{\tilde{\alpha}^{(L)}(\mathbf{z})\geq 0\}}W^{(L+1)T}$$

Recalling that $\tilde{\alpha}^{(l)} = W^{(l)}\tilde{\alpha}^{(l-1)}$ and applying Lemma D.3 on the the term $A$ above, we obtain

$$\langle \delta^{(l)}(\mathbf{z}), \delta^{(l)}(\mathbf{x})\rangle = \frac{\pi - \theta^{(l-1)}(\mathbf{x}, \mathbf{z})}{\pi}\langle \delta^{(l+1)}(\mathbf{z}), \delta^{(l+1)}(\mathbf{x})\rangle.$$

Recursively applying the above formula for $l' = l, l+1, \cdots, L$, and noticing that $\delta^{(L+1)} = 1$, we have

$$\langle \delta^{(l)}(\mathbf{z}), \delta^{(l)}(\mathbf{x})\rangle = \prod_{l'=l-1}^{L+1}\left(1 - \frac{\theta^{(l')}(\mathbf{x}, \mathbf{z})}{\pi}\right). \quad (34)$$

Combining Eq.(32), (33) and (34), we have

$$\langle \nabla_l f(\mathbf{z}), \nabla_l f(\mathbf{x})\rangle = \|\mathbf{x}\|\|\mathbf{z}\|\cos\theta^{(l-1)}(\mathbf{x}, \mathbf{z})\prod_{l'=l-1}^{L-1}\left(1 - \frac{\theta^{(l')}(\mathbf{x}, \mathbf{z})}{\pi}\right). \quad (35)$$

For the inner product between the full model gradients, we have

$$\langle \nabla f(\mathbf{z}), \nabla f(\mathbf{x})\rangle = \sum_{l=1}^{L+1}\langle \nabla_l f(\mathbf{z}), \nabla_l f(\mathbf{x})\rangle = \|\mathbf{x}\|\|\mathbf{z}\|\sum_{l=0}^{L}\left[\cos\theta^{(l)}(\mathbf{x}, \mathbf{z})\prod_{l'=l}^{L-1}\left(1 - \frac{\theta^{(l')}(\mathbf{x}, \mathbf{z})}{\pi}\right)\right]. \quad (36)$$

Putting $\mathbf{x} = \mathbf{z}$ in the above equation, we have $\theta^{(l)}(\mathbf{x}, \mathbf{z}) = 0$ for all $l \in [L]$, and obtain

$$\|\nabla f(\mathbf{x})\|^2 = \|\mathbf{x}\|^2 \cdot (L+1). \quad (37)$$

Hence, we have

$$\cos\phi(\mathbf{x}, \mathbf{z}) = \frac{\langle \nabla f(\mathbf{z}), \nabla f(\mathbf{x})\rangle}{\|\nabla f(\mathbf{x})\|\|\nabla f(\mathbf{z})\|} = \frac{1}{L+1}\sum_{l=0}^{L}\left[\cos\theta^{(l)}(\mathbf{x}, \mathbf{z})\prod_{l'=l}^{L-1}(1 - \theta^{(l')}(\mathbf{x}, \mathbf{z})/\pi)\right]. \quad (38)$$

$\square$

---

[1]With a bit of abuse of notation, we refer to the flattened vectors of $\nabla_l f$ in the inner product.

### D.5  PROOF OF THEOREM 4.4

*Proof.* For simplicity of notation, we don't explicitly write out the dependent on the inputs $\mathbf{x}, \mathbf{z}$, and write $\theta^{(l)} \triangleq \theta^{(l)}(\mathbf{x}, \mathbf{z})$, and $\phi \triangleq \phi(\mathbf{x}, \mathbf{z})$. We start the proof with the relation provided by Lemma 4.3.

$$
\cos\phi(\mathbf{x}, \mathbf{z}) = \frac{1}{L+1}\sum_{l=0}^{L}\left[\cos\theta^{(l)}\prod_{l'=l}^{L-1}(1-\theta^{(l')}/\pi)\right]
$$

$$
\overset{(a)}{=} \frac{1}{L+1}\sum_{l=0}^{L}\left[\cos\theta^{(0)}\prod_{l'=0}^{l-1}\left(1+\frac{1}{\pi}\tan\theta^{(l')}-\frac{1}{\pi}\theta^{(l')}\right)\prod_{l'=l}^{L-1}(1-\theta^{(l')}/\pi)\right]
$$

$$
\overset{(b)}{=} \frac{1}{L+1}\sum_{l=0}^{L}\left[\cos\theta^{(0)}\prod_{l'=0}^{l-1}\left(1+\frac{1}{3\pi}(\theta^{(l')})^3+o(\theta^{(l')})^3\right)\prod_{l'=l}^{L-1}(1-\theta^{(l')}/\pi)\right]
$$

$$
\overset{(c)}{=} \frac{\cos\theta^{(0)}}{L+1}\sum_{l=0}^{L}\left[\prod_{l'=0}^{l-1}\left(1+\frac{1}{3\pi}(\theta^{(0)})^3+o(\theta^{(0)})^3\right)\right.
$$
$$
\left.\times\prod_{l'=l}^{L-1}\left(1-\frac{1}{\pi}\theta^{(0)}+\frac{l'}{3\pi^2}(\theta^{(0)})^2+o((\theta^{(0)})^2)\right)\right]
$$

$$
= \frac{\cos\theta^{(0)}}{L+1}\sum_{l=0}^{L}\left(1-\frac{L-l}{\pi}\theta^{(0)}+\frac{(L-l)(2L-l-2)}{3\pi^2}(\theta^{(0)})^2+o((\theta^{(0)})^2)\right)
$$

$$
= \cos\theta^{(0)}\left(1-\frac{L}{2}\theta^{(0)}+o(\theta^{(0)})\right).
$$

$\square$

### D.6  PROOF OF THEOREM 5.3

*Proof.* For this shallow ReLU network, the model gradient, for an arbitrary input $\mathbf{x}$, is written as

$$
\nabla f(\mathbf{x}) = \mathbf{x}\delta(\mathbf{x}) \in \mathbb{R}^{d\times m}, \tag{39}
$$

where $\delta(\mathbf{x}) \in \mathbb{R}^{1\times m}$ has the following expression

$$
\delta(\mathbf{x}) = \sqrt{\frac{2}{m}}\mathbf{v}^T\mathbb{I}_{\{W\mathbf{x}\geq 0\}}.
$$

At initialization, $W$ is a random matrix. Utilizing Lemma D.3, it is easy to check that $\|\delta(\mathbf{x})\| = 1$ for all input $\mathbf{x}$ in the infinite width limit.

Recall that the NTK $K = FF^T$, where the gradient feature matrix $F$ consist of the gradient feature vectors $\nabla f(\mathbf{x})$ for all $\mathbf{x}$ for the dataset. Hence, the smallest eigenvalue $\lambda_{min}(K)$ satisfies

$$
\lambda_{min}(K) = \min_{\mathbf{u}\neq 0}\frac{\mathbf{u}^T K\mathbf{u}}{\|\mathbf{u}\|^2} = \min_{\mathbf{u}\neq 0}\frac{\|\sum_{i=1}^{n}u_i\nabla f(\mathbf{x}_i)\|^2}{\sum_{i=1}^{n}u_i^2}
$$

$$
= \min_{\mathbf{u}\neq 0}\frac{\sum_{j=1}^{m}\|\sum_{i=1}^{n}u_i\delta_j(\mathbf{x}_i)\mathbf{x}_i\|^2}{\sum_{i=1}^{n}u_i^2}
$$

$$
= \min_{\mathbf{u}\neq 0}\sum_{j=1}^{m}\frac{\sum_{i=1}^{n}(u_i\delta_j(\mathbf{x}_i))^2}{\sum_{i=1}^{n}u_i^2}\frac{\|\sum_{i=1}^{n}u_i\delta_j(\mathbf{x}_i)\mathbf{x}_i\|^2}{\sum_{i=1}^{n}(u_i\delta_j(\mathbf{x}_i))^2}
$$

$$
\overset{(a)}{>} \min_{\mathbf{u}\neq 0}\sum_{j=1}^{m}\frac{\sum_{i=1}^{n}(u_i\delta_j(\mathbf{x}_i))^2}{\sum_{i=1}^{n}u_i^2}\lambda_{min}(G). \tag{40}
$$

In the inequality $(a)$ above, we made the following treatment: for each fixed $j$, we consider $u_i\delta_j(\mathbf{x}_i)$ as the $i$-th component of a vector $\mathbf{u}'_j$; by definition, the minimum eigenvalue of Gram matrix

$$
\lambda_{min}(G) = \min_{\mathbf{u}'\neq 0}(\mathbf{u}')^T G\mathbf{u}'/\|\mathbf{u}'\|^2 \leq (\mathbf{u}'_j)^T G\mathbf{u}'_j/\|\mathbf{u}'_j\|^2, \ \ \forall j; \tag{41}
$$

moreover, this $\leq$ inequality becomes equality, if and only if all $\mathbf{u}'_j$ are the same and equal to $\arg\min_{\mathbf{u}'\neq 0}(\mathbf{u}')^T G \mathbf{u}'/\|\mathbf{u}'\|^2$. As easy to see, when the dataset is not degenerate, for different $j$, $\mathbf{u}'_j$ are different, hence only the strict inequality $<$ holds in step $(a)$.

Continuing from Eq.(40), we have

$$\lambda_{min}(K) > \min_{\mathbf{u}\neq 0}\sum_{j=1}^m \frac{\sum_{i=1}^n (u_i \delta_j(\mathbf{x}_i))^2}{\sum_{i=1}^n u_i^2}\lambda_{min}(G)$$

$$= \min_{\mathbf{u}\neq 0}\frac{\sum_{i=1}^n u_i^2\|\delta(\mathbf{x}_i)\|^2}{\sum_{i=1}^n u_i^2}\lambda_{min}(G)$$

$$= \min_{\mathbf{u}\neq 0}\frac{\sum_{i=1}^n u_i^2}{\sum_{i=1}^n u_i^2}\lambda_{min}(G) = \lambda_{min}(G).$$

Therefore, we showed that $\lambda_{min}(K) > \lambda_{min}(G)$.

As for the largest eigenvalue $\lambda_{max}(K)$, we can apply the same logic above for $\lambda_{min}(K)$ (except replacing the $\min$ operator by $\max$ and have $<$ in step $(a)$) to get $\lambda_{max}(K) > \lambda_{max}(G)$.

Therefore, by definition of condition number, the condition number $\kappa$ of NTK is strictly smaller than the Gram matrix condition number $\kappa_0$. $\qquad\square$

### D.7   PROOF OF THEOREM 5.2

*Proof.* According to the definition of NTK and Lemma 4.3, the NTK matrix $K$ for this dataset $\mathcal{D} = \{(\mathbf{x}_1, y_1), (\mathbf{x}_2, y_2)\}$ is (NTK is normalized by the factor $1/(L+1)^2$):

$$K = \begin{pmatrix} \|\nabla f(\mathbf{x}_1)\|^2 & \langle\nabla f(\mathbf{x}_1), \nabla f(\mathbf{x}_2)\rangle \\ \langle\nabla f(\mathbf{x}_2), \nabla f(\mathbf{x}_1)\rangle & \|\nabla f(\mathbf{x}_2)\|^2 \end{pmatrix} = \begin{pmatrix} \|\mathbf{x}_1\|^2 & \|\mathbf{x}_1\|\|\mathbf{x}_2\|\cos\phi \\ \|\mathbf{x}_1\|\|\mathbf{x}_2\|\cos\phi & \|\mathbf{x}_2\|^2 \end{pmatrix}.$$

The eigenvalues of the NTK matrix $K$ are given by

$$\lambda_1(K) = \frac{1}{2}\left(\|\mathbf{x}_1\|^2 + \|\mathbf{x}_2\|^2 + \sqrt{\|\mathbf{x}_1\|^4 + \|\mathbf{x}_2\|^4 + \|\mathbf{x}_1\|^2\|\mathbf{x}_2\|^2\cos 2\phi}\right), \qquad (42a)$$

$$\lambda_2(K) = \frac{1}{2}\left(\|\mathbf{x}_1\|^2 + \|\mathbf{x}_2\|^2 - \sqrt{\|\mathbf{x}_1\|^4 + \|\mathbf{x}_2\|^4 + \|\mathbf{x}_1\|^2\|\mathbf{x}_2\|^2\cos 2\phi}\right). \qquad (42b)$$

Similarly, for the Gram matrix $G$, we have

$$G = \begin{pmatrix} \|\mathbf{x}_1\|^2 & \mathbf{x}_1^T\mathbf{x}_2 \\ \mathbf{x}_1^T\mathbf{x}_2 & \|\mathbf{x}_2\|^2 \end{pmatrix} = \begin{pmatrix} \|\mathbf{x}_1\|^2 & \|\mathbf{x}_1\|\|\mathbf{x}_2\|\cos\theta_{in} \\ \|\mathbf{x}_1\|\|\mathbf{x}_2\|\cos\theta_{in} & \|\mathbf{x}_2\|^2 \end{pmatrix},$$

and its eigenvalues as

$$\lambda_1(G) = \frac{1}{2}\left(\|\mathbf{x}_1\|^2 + \|\mathbf{x}_2\|^2 + \sqrt{\|\mathbf{x}_1\|^4 + \|\mathbf{x}_2\|^4 + \|\mathbf{x}_1\|^2\|\mathbf{x}_2\|^2\cos 2\theta_{in}}\right),$$

$$\lambda_2(G) = \frac{1}{2}\left(\|\mathbf{x}_1\|^2 + \|\mathbf{x}_2\|^2 - \sqrt{\|\mathbf{x}_1\|^4 + \|\mathbf{x}_2\|^4 + \|\mathbf{x}_1\|^2\|\mathbf{x}_2\|^2\cos 2\theta_{in}}\right).$$

By Theorem 4.4, we have $\cos\phi < \cos\theta_{in}$, when $\theta_{in} = o(1/L)$ and $\theta_{in} \neq 0$. Hence, we have the following relations

$$\lambda_1(G) > \lambda_1(K) > \lambda_2(K) > \lambda_2(G),$$

which immediately implies $\kappa < \kappa_0$.

When comparing ReLU networks with different depths, i.e., network $f_1$ with depth $L_1$ and network $f_2$ with depth $L_2$ with $L_1 > L_2$, notice that in Eq.(42) the top eigenvalue $\lambda_1$ monotonically decreases in $\phi$, and the bottom (smaller) eigenvalue $\lambda_2$ monotonically increases in $\phi$. By Theorem 4.4, we know that the deeper ReLU network $f_1$ has a better data separation than the shallower one $f_2$, i.e., $\phi_{f_1} > \phi_{f_2}$. Hence, we get

$$\lambda_1(K_{f_2}) > \lambda_1(K_{f_1}) > \lambda_2(K_{f_1}) > \lambda_2(K_{f_2}). \qquad (43)$$

Therefore, we obtain $\kappa_{f_1} < \kappa_{f_2}$. Namely the deeper ReLU network has a smaller NTK condition number. $\qquad\square$

# E  TECHNICAL PROOFS

## E.1  PROOF OF LEMMA C.1

*Proof.* We denote $A_{ij}$ as the $(i, j)$-th entry of the matrix $A$. Therefore, $(A^T A)_{ij} = \sum_{k=1}^{m} A_{ki} A_{kj}$. First we find the mean of each $(A^T A)_{ij}$. Since $A_{ij}$ are i.i.d. and has zero mean, we can easily see that for any index $k$,

$$\mathbb{E}[A_{ki} A_{kj}] = \begin{cases} 1, & \text{if } i = j \\ 0, & \text{otherwise} \end{cases}.$$

Consequently,

$$\mathbb{E}[(\frac{1}{m} A^T A)_{ij}] = \begin{cases} 1, & \text{if } i = j \\ 0, & \text{otherwise} \end{cases}.$$

That is $\mathbb{E}[\frac{1}{m} A^T A] = I_d$.

Now we consider the variance of each $(A^T A)_{ij}$. If $i \neq j$ we can explicitly write,

$$
\begin{aligned}
Var\left[\frac{1}{m}(A^T A)_{ij}\right] &= \frac{1}{m^2} \cdot \mathbb{E}\left[\sum_{k_1=1}^{m} \sum_{k_2=1}^{m} A_{k_1 i} A_{k_1 j} A_{k_2 i} A_{k_2 j}\right] \\
&= \frac{1}{m^2} \cdot \sum_{k_1=1}^{m} \sum_{k_2=1}^{m} \mathbb{E}\left[A_{k_1 i} A_{k_1 j} A_{k_2 i} A_{k_2 j}\right] \\
&= \frac{1}{m^2} \left(\sum_{k=1}^{m} \mathbb{E}\left[A_{ki}^2 A_{kj}^2\right] + \sum_{k_1 \neq k_2} \mathbb{E}\left[A_{k_1 i} A_{k_1 j} A_{k_2 i} A_{k_2 j}\right]\right) \\
&= \frac{1}{m^2} \left(\sum_{k=1}^{m} \mathbb{E}\left[A_{ki}^2\right] \mathbb{E}\left[A_{kj}^2\right] + \sum_{k_1 \neq k_2} \mathbb{E}[A_{k_1 i}] \mathbb{E}[A_{k_1 j}] \mathbb{E}[A_{k_2 i}] \mathbb{E}[A_{k_2 j}]\right) \\
&= \frac{1}{m^2} \cdot (m + 0) = \frac{1}{m}.
\end{aligned}
$$

In the case of $i = j$, then,

$$Var\left[\frac{1}{m}(A^T A)_{ii}\right] = \frac{1}{m^2} \cdot Var\left[\sum_{k=1}^{m} A_{ki}^2\right] = \frac{1}{m^2} \cdot \sum_{k=1}^{m} Var\left[A_{ki}^2\right] \stackrel{(a)}{=} \frac{1}{m^2}(m \cdot 2) = \frac{2}{m}. \quad (44)$$

In the equality (a) above, we used the fact that $A_{ki}^2 \sim \chi^2(1)$. Therefore, $\lim_{m \to \infty} Var(\frac{1}{m}(A^T A)) = 0$.

Now applying Chebyshev's inequality we get,

$$Pr(|\frac{1}{m} A^T A - I_d| \geq \epsilon) \leq \frac{Var(\frac{1}{m}(A^T A))}{\epsilon} \quad (45)$$

Obviously for any $\epsilon \geq 0$ as $m \to \infty$, the R.H.S. goes to zero. Thus, $\frac{1}{m} A^T A \to I_{d \times d}$, in probability. $\square$

## E.2  PROOF OF LEMMA D.1

*Proof.* Note that the random vector $\mathbf{w}$ is isotropically distributed and that only inner products $\mathbf{w}^T \mathbf{v}_1$ and $\mathbf{w}^T \mathbf{v}_2$ appear, hence we can assume without loss of generality that (if not, one can rotate the coordinate system to make it true):

$$
\begin{aligned}
\mathbf{v}_1 &= \|\mathbf{v}_1\|(1, 0, 0, \cdots, 0), \\
\mathbf{v}_2 &= \|\mathbf{v}_2\|(\cos\theta, \sin\theta, 0, \cdots, 0).
\end{aligned}
$$

In this setting, the only relevant parts of $\mathbf{w}$ are its first two scalar components $w_1$ and $w_2$. Define $\tilde{\mathbf{w}}$ as

$$\tilde{\mathbf{w}} = (w_1, w_2, 0, \cdots, 0) = \sqrt{w_1^2 + w_2^2}(\cos\omega, \sin\omega, 0, \cdots, 0). \tag{46}$$

Then,

$$\mathbb{P}[(\mathbf{w}^T\mathbf{v}_1 \geq 0) \wedge (\mathbf{w}^T\mathbf{v}_2 \geq 0)] = \mathbb{P}[(\tilde{\mathbf{w}}^T\mathbf{v}_1 \geq 0) \wedge (\tilde{\mathbf{w}}^T\mathbf{v}_2 \geq 0)] = \frac{1}{2\pi}\int_{\theta-\frac{\pi}{2}}^{\frac{\pi}{2}} d\omega = \frac{1}{2} - \frac{\theta}{2\pi}.$$

$\square$

### E.3  PROOF OF LEMMA D.2

*Proof.* Note that the ReLU activation function $\sigma(z)$ can be written as $z\mathbb{I}_{z\geq 0}$. We have,

$$\langle \mathbf{u}_1, \mathbf{u}_2 \rangle = \frac{2}{q}\mathbf{v}_1^T W^T \mathbb{I}_{\{W\mathbf{v}_1\geq 0, W\mathbf{v}_2\geq 0\}} W\mathbf{v}_2$$

$$= \frac{2}{q}\sum_{i=1}^{q}\mathbf{v}_1^T(W_{i\cdot})^T \mathbb{I}_{\{W_{i\cdot}\mathbf{v}_1\geq 0, W_{i\cdot}\mathbf{v}_2\geq 0\}} W_{i\cdot}\mathbf{v}_2$$

$$\overset{q\to\infty}{=} 2\mathbb{E}_{\mathbf{w}\sim\mathcal{N}(0,I_{p\times p})}[\mathbf{v}_1^T\mathbf{w}\mathbb{I}_{\{\mathbf{w}^T\mathbf{v}_1\geq 0, \mathbf{w}^T\mathbf{v}_2\geq 0\}}\mathbf{w}^T\mathbf{v}_2]$$

Note that the random vector $\mathbf{w}$ is isotropically distributed and that only inner products $\mathbf{w}^T\mathbf{v}_1$ and $\mathbf{w}^T\mathbf{v}_2$ appear, hence we can assume without loss of generality that (if not, one can rotate the coordinate system to make it true):

$$\mathbf{v}_1 = \|\mathbf{v}_1\|(1, 0, 0, \cdots, 0),$$
$$\mathbf{v}_2 = \|\mathbf{v}_2\|(\cos\theta, \sin\theta, 0, \cdots, 0).$$

In this setting, the only relevant parts of $\mathbf{w}$ are its first two scalar components $w_1$ and $w_2$. Define $\tilde{\mathbf{w}}$ as

$$\tilde{\mathbf{w}} = (w_1, w_2, 0, \cdots, 0) = \sqrt{w_1^2 + w_2^2}(\cos\omega, \sin\omega, 0, \cdots, 0). \tag{47}$$

Then, in the limit of $q \to \infty$,

$$\langle \mathbf{u}_1, \mathbf{u}_2 \rangle = 2\mathbb{E}_{\mathbf{w}\sim\mathcal{N}(0,I_{p\times p})}[\mathbf{v}_1^T\mathbf{w}\mathbb{I}_{\{\mathbf{w}^T\mathbf{v}_1\geq 0, \mathbf{w}^T\mathbf{v}_2\geq 0\}}\mathbf{w}^T\mathbf{v}_2]$$

$$= 2\mathbb{E}_{\tilde{\mathbf{w}}\sim\mathcal{N}(0,I_{2\times 2})}[\mathbf{v}_1^T\tilde{\mathbf{w}}\mathbb{I}_{\{\tilde{\mathbf{w}}^T\mathbf{v}_1\geq 0, \tilde{\mathbf{w}}^T\mathbf{v}_2\geq 0\}}\tilde{\mathbf{w}}^T\mathbf{v}_2]$$

$$= 2\|\mathbf{v}_1\|\|\mathbf{v}_2\| \cdot \mathbb{E}_{\tilde{\mathbf{w}}\sim\mathcal{N}(0,I_{2\times 2})}[\|\tilde{\mathbf{w}}\|^2] \cdot \frac{1}{2\pi}\int_{\theta-\frac{\pi}{2}}^{\frac{\pi}{2}}\cos\omega\cos(\theta-\omega)d\omega$$

$$= 2\|\mathbf{v}_1\|\|\mathbf{v}_2\| \cdot 2 \cdot \frac{1}{4\pi}\left((\pi-\theta)\cos\theta + \sin\theta\right)$$

$$= \|\mathbf{v}_1\|\|\mathbf{v}_2\|\frac{1}{\pi}\left((\pi-\theta)\cos\theta + \sin\theta\right).$$

$\square$

### E.4  PROOF OF LEMMA D.3

*Proof.*

$$A_1 A_2^T = \frac{2}{q}\sum_{k=1}^{q}U_{\cdot k}\mathbb{I}_{\{W_{k\cdot}\mathbf{v}_1\geq 0, W_{k\cdot}\mathbf{v}_2\geq 0\}}(U_{\cdot k})^T$$

$$\overset{q\to\infty}{=} 2 \cdot \mathbb{E}_{\mathbf{u}\sim\mathcal{N}(0,I_{s\times s}), \mathbf{w}\sim\mathcal{N}(0,I_{p\times p})}[\mathbf{u}\mathbf{u}^T\mathbb{I}_{\{\mathbf{w}^T\mathbf{v}_1\geq 0, \mathbf{w}^T\mathbf{v}_2\geq 0\}}]$$

$$\overset{(a)}{=} 2 \cdot \mathbb{E}_{\mathbf{u}\sim\mathcal{N}(0,I_{s\times s})}[\mathbf{u}\mathbf{u}^T] \cdot \mathbb{E}_{\mathbf{w}\sim\mathcal{N}(0,I_{p\times p})}[\mathbb{I}_{\{\mathbf{w}^T\mathbf{v}_1\geq 0, \mathbf{w}^T\mathbf{v}_2\geq 0\}}]$$

$$= 2 \cdot \mathbb{E}_{\mathbf{u}\sim\mathcal{N}(0,I_{s\times s})}[\mathbf{u}\mathbf{u}^T] \cdot \mathbb{P}[(\mathbf{w}^T\mathbf{v}_1 \geq 0) \wedge (\mathbf{w}^T\mathbf{v}_2 \geq 0)]$$

$$\overset{(b)}{=} \frac{\pi-\theta}{\pi}I_{s\times s}.$$

In the step (a) above, we used the fact that $U$ is independent of $W$, $\mathbf{v}_1$ and $\mathbf{v}_2$. In the step (b) above, we applied Lemma D.1, and used the fact that $\mathbb{E}_{\mathbf{u}\sim\mathcal{N}(0,I_{s\times s})}[\mathbf{u}\mathbf{u}^T] = I_{s\times s}$. $\square$

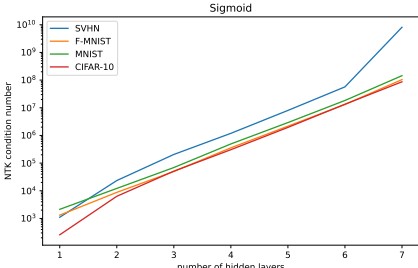 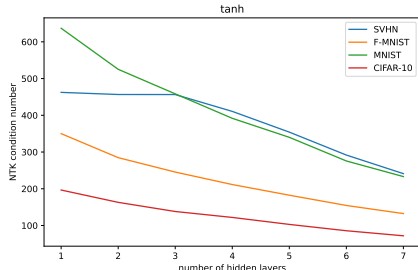

Figure 5: NTK condition number vs. depth, for *sigmoid*-activated network and *tanh*-activated network.

## F NUMERICAL RESULTS OF OTHER ACTIVATION FUNCTIONS

In this section, we show some preliminary numerical results of some other non-linear activation functions, although the main focus of this paper is ReLU.

Specifically, analogous to what we did for ReLU network, we compute the NTK condition number for the following two types of non-linearly activated neural networks at random initialization: *sigmoid*-activated network and *tanh*-activated network. In both cases, we use the same network width, 512, as in Figure 2 for ReLU network. The scaling factor, $\sqrt{2/m_l}$ in Eq.(1), was replaced by $\sqrt{c_\sigma/m_l}$, where $c_\sigma$ is a activation-specific constant and is defined as $c_\sigma = \left(\mathbb{E}_{x\sim\mathcal{N}(0,1)}[\sigma(x)^2]\right)^{-1}$ (see for example Eq.(2) of Du et al. (2019a)).

Figure 5 shows the dependence of the NTK condition number on the network depth. We observe that different non-linear activation function may have different effects on the NTK condition numbers $\kappa$. As the figure tells, *tanh* also helps to decrease the condition number (similar to ReLU), while *sigmoid* has the opposite effect, worsening the NTK conditioning.

A theoretical analysis of these non-linear activation functions are out of the scope of this paper, but we expect future work will theoretically clarify the exact effects of different types of non-linear activation functions.