# OpenReview forum: "ReLU soothes NTK conditioning and accelerates optimization for wide neural networks"
_ICLR.cc/2024/Conference — Submitted to ICLR 2024_

### Official Review · Reviewer_XQe2 · 2023-10-30

**Soundness:** 3 good
**Presentation:** 3 good
**Contribution:** 1 poor
**Rating:** 3
**Confidence:** 4

**Summary:**

This paper studies the effect of the ReLU activation function in terms of (i) the data separation in the feature space of the model gradient, and (ii) the conditioning of the NTK. As for (i), Theorem 4.4 proves that, if two network inputs have small angle, then the model gradients become less and less aligned as the depth of the network increases. As for (ii), Theorems 5.2 and 5.3 show that the condition number of the NTK is smaller than that of the Gram matrix containing the inputs (meaning that the NTK is better conditioned). Specifically, Theorem 5.2 looks at the case in which we have only 2 data samples and shows that the condition number decreases with depth; Theorem 5.3 looks at an arbitrary non-degenerate dataset (i.e., inputs not aligned), considers the NTK of a two-layer network where the outer layer is not trained, and proves that the condition number of such NTK is smaller than the condition number of the Gram matrix of the input. All NTK results refer to the infinite width limit. For comparison, in a linear network, data separation and condition number of the NTK do not change with depth. The fact that the NTK is better conditioned has implications on optimization which are shown via numerical results on MNIST, f-MNIST and Librispeech (see Section 6). Numerical experiments also demonstrate the better separation and conditioning of ReLU networks (compared to linear networks).

**Strengths:**

* While there has been some work on the impact of the activation function on the NTK (mentioned below), the results presented here are new. Specifically, the authors focus on the regime in which the angle between two data points is small and establish what's the effect of the ReLU nonlinearity on the corresponding gradients in the infinite-width limit.

* The numerical results show a similar phenomenology also at finite widths, which is a nice addition.

* The results appear correct (after also looking at the appendix).

**Weaknesses:**

Overall, although the results are correct and the regime being investigated is new, the findings are a bit underwhelming, due to the restrictive regime in which they hold.

Specifically, Theorem 4.4 only tracks the input angle which is assumed to be $o(1/L)$. Other relevant parameters such as the input dimension $d$ and the number of samples $n$ are assumed to be constant (which is rather unrealistic in typical datasets).

The regime being restrictive is even more evident in the NTK results. Theorem 5.2 holds only for two data points. Theorem 5.3 holds for a general two layer network with the outer layer being fixed. However, it implicitly requires that the input dimension $d$ is bigger than the number of samples $n$. In fact, if that's not the case, $G$ is not full rank, and the statement becomes trivial (as the smallest eigenvalue of $G$ is $0$, and it is well known that the smallest eigenvalue of the NTK is bounded away from $0$). Note that having $d>n$ is violated in all the experiments of Section 5. Actually, the numbers reported in Figure 2(b) when $L=0$ are a bit suspicious. I would expect the condition number to be $\infty$ since $G$ has at most rank $d$. Or am I missing something here?

**Questions:**

(1) Can the authors comment on the points reported in Figure 2(b) when $L=0$?

(2) A clear way in which the results can be made stronger is to track the quantities $d, n$ in the various results. Having some assumption on the data (e.g., sub-Gaussian tails) may be necessary in order to provide non-trivial statements. Also being able to track the number of neurons $m$ (and therefore consider the empirical NTK) would add value to the narrative of the paper.

(3) There are some works that study the impact of the activation function on the NTK, see [R1], [R2]. How do the results of the manuscript compare to such existing works?

[R1] Panighahi et al., "Effect of Activation Functions on the Training of Overparametrized Neural Nets", ICLR 2020.

[R2] Zhu et al., "Generalization Properties of NAS under Activation and Skip Connection Search", NeurIPS 2022.

---

> ### Author Response · Authors · 2023-11-17
> **Response to Reviewer XQe2: part I**
>
> We thank the reviewer for the constructive comments. We are grateful that the reviewer recognized the novelty of the work. We also understand your concerns. We would like to address your concerns and questions one by one below.
>
> **W1:** *“... However, it implicitly requires that the input dimension $d$ is bigger than the number of samples $n$. In fact, if that is not the case, $G$ is not full rank, and the statement becomes trivial …”*
>
> **A:** We apologize for not making this point clear in the submission. **In short, we do not require $d>n$.** Technically speaking, it is the “*effective*” NTK condition number (largest eigenvalue divided by the smallest *non-zero* eigenvalue) that controls the convergence rate. When $G$ is not full rank, this relates to the smallest *non-zero* eigenvalue of $G$. We would like to explain this in detail below:
>
> Let’s consider the case $d<n$, where the concern arises. *Why it is the smallest *non-zero* eigenvalue?*
>
> – In this case, the Gram matrix $G$ is just the NTK of the linear model, and is not full rank. Note that this linear model is in the *under-parameterized regime*. As we know, for the linear model, the NTK $K=XX^T$ has the same spectrum as the Hessian of least square loss $H=X^TX$, except the zero eigenvalues. This Hessian is expected to have full rank, and the least square loss is convex. As is well-known, the condition number is defined as $\lambda_{max}(H)/\lambda_{min}(H)$, which is equivalent to $\lambda_{max}(K)/\lambda_{min}^*(K)$, with $\lambda_{min}^*(K)$ as the smallest **non-zero** eigenvalue of NTK.
>
> – How about the linear network and ReLU network? In the infinite width limit, they are essentially linear models, with the model gradient $\nabla f(x)$ (instead of original input $x$) as the feature. These linear models are *over-parameterized*, and have a hyper-plane as the solution set $S$. Intuitively, the optimization (by gradient descent) occurs in sub-spaces that are perpendicular to $S$. In addition, the zero eigenspace of NTK are also perpendicular to $S$. Hence, the zero eigenvalues of NTK contribute nothing to the optimization. Therefore, the condition number of optimizational interest is the “effective” condition number (i.e., ignoring the zero eigenvalues).
>
> In summary, our results do *not* require $d>n$. Our results essentially rely on the “*effective*” condition number of NTK. We will clarify this point in the revision.
>
> **W2:** *“Theorem 5.3 holds for a general two layer network with the outer layer being fixed”.*
>
> **A:** This setting is commonly used in literature, including very good papers, for example Simon S Du, Xiyu Zhai, Barnabas Poczos, and Aarti Singh. “Gradient descent provably optimizes over-parameterized neural networks”. We also observe that many novel findings originally come out in simple settings, which does not affect its completion by later works. We believe that this simple setting is a good starting point. We are aware that for deeper networks it requires much complicated analysis. It is believable the same results hold for deep networks, as partially evident by Figure 2 and Theorem 5.2.
>
>
> **W3:** *“the numbers reported in Figure 2(b) when $L=0$ are a bit suspicious” & Q1: “Can the authors comment on the points reported in Figure 2(b) when $L=0$?”*
>
> **A:** In the experiment to compute the numbers in Figure 2(b), we actually evaluated NTK based on batches of size $512$. This is due to the limitation by our computational resource, which could not compute and store larger Jacobian matrices. The reported numbers are averaged over the batches. In this case, the Gram matrix $G$ is always full rank, hence it is not surprising that $\kappa$ is finite. One can anticipate that the condition numbers $\kappa$ at $L=0$ may increase with larger batch size, and perhaps be infinite if it exceeds $d$. We will clarify this setting in the revision.

---

> > ### Author Response · Authors · 2023-11-17
> > **Response to Reviewer XQe2: part II**
> >
> > **Q3:** *“There are some works that study the impact of the activation function on the NTK, see [R1], [R2]. How do the results of the manuscript compare to such existing works?”*
> >
> > **A:** The most relevant part of [R1] and [R2] is their theoretical bounds on the smallest eigenvalue $\lambda_{min}$ of NTK. [R1] provides an $\Omega$-type bound on $\lambda_{min}$ (in their Theorem 4.1 and 4.2). [R2] gives a lower-bound on $\lambda_{min}$ by Hermite coefficient (see Theorem 1 therein). In addition, both work studied the dependence of the lower-bound on the activation type. These results are surely interesting, we will cite them in the revision.
> > However, based on these analyses, it is hard to judge whether any of these non-linear activations helps to decrease the condition number, because we are not sure about the tightness of the lower-bound. (Although [R2] provides an upper bound, it seems loose, for example for ReLU network, the upper bound for $\lambda_{min}$ is the number of layers $L$, see Theorem 1). On the other hand, we directly compare the condition numbers, not their bounds.
> >
> > In this paper, we showed a new effect for certain non-linear activations – decreasing the condition number of NTK. Before this, non-linear activations were known to increase the expressivity. We view this result as an addition to our understanding of the effect of non-linear activation functions. We don’t see similar results in either [R1] or [R2].

---

> > > ### Comment · Reviewer_XQe2 · 2023-11-20
> > > **thanks for the response, but some issues remain**
> > >
> > > I would like to thank the authors for the thoughtful response. However, I disagree with the following statements:
> > >
> > > "Intuitively, the optimization (by gradient descent) occurs in sub-spaces that are perpendicular to $S$. In addition, the zero eigenspace of NTK are also perpendicular to $S$. Hence, the zero eigenvalues of NTK contribute nothing to the optimization. Therefore, the condition number of optimizational interest is the “effective” condition number (i.e., ignoring the zero eigenvalues)."
> > >
> > > If the NTK is low rank, then gradient descent can be stuck precisely in the span of the eigenvectors corresponding to the 0 eigenvalues. In fact, all the works proving convergence of gradient descent in the NTK regime either assume or prove that the smallest NTK eigenvalue is bounded away from 0 (including the paper by Du et al. cited by the authors, see their Theorem 3.1). Note also that, if we were to consider the "effective" NTK condition number as suggested by the authors, then one could just set to 0 the value of the smallest non-zero eigenvalue. The resulting matrix is a lower bound on the NTK (in the PSD sense), but the condition number has improved.
> > >
> > > For this reason, I don't quite understand the motivation (from an optimization viewpoint) of removing the 0 eigenvalues from the NTK.

---

> ### Author Response · Authors · 2023-11-21
> **thanks for the follow up question**
>
> We would like to thank the reviewer for the follow up question. Sorry for the confusion.
>
> Please note that the NTK (a $n\times n$ matrix) and loss Hessian (a $p\times p$ matrix) have different spaces, where $p$ is the number of parameters which is much larger than $n$ in the over-parameterized networks. Also note that it is this $p$-dimensional parameter space that the weights are updated and gradient descent resides in. Hence, we don't quite see how "If the NTK is low rank, then gradient descent can be stuck precisely in the span of the eigenvectors corresponding to the 0 eigenvalues", as gradient descent is not in the same space as the eigenvectors.
>
> For ReLU network, the NTK is indeed of full rank, as shown by Du et al. However, no matter NTK is of full rank or not. As $p\gg n$ for over-parameterized models, the loss Hessian, $p\times p$, is low rank and have a lot of zero eigenvalues. Our statement that you quoted is actually talking about the gradient descent in this $p$-dimensional parameter space.
>
> We hope this can address your concern. We are also happy to discuss if you have further questions.

---

> > ### Comment · Reviewer_XQe2 · 2023-11-22
> > **discussion**
> >
> > I perfectly agree with the authors in that the NTK is an $n$ by $n$ matrix, while gradient descent acts on a $p$-dimensional space. However, the smallest eigenvalue of this $n$ by $n$ NTK is still what matters for the analysis of gradient descent in the NTK regime. The requirement that this eigenvalue is bounded away from 0 is present in pretty much any paper that does a convergence proof in the NTK regime. I can quote as a representative example Theorem 5.1 in the review paper https://arxiv.org/pdf/2103.09177.pdf
> >
> > In short, I still fail to see the importance of the 'effective' condition number. As mentioned above, I can always push to 0 the smallest non-zero eigenvalue, get a lower bound on the NTK in the PSD sense and improve the effective condition number.

---

> > > ### Author Response · Authors · 2023-11-22
> > > **reply to "discussion"**
> > >
> > > Thanks for pointing this out. You are correct. We agree that, in the case of Gram matrix $G$ that is not full rank, the “effective” condition number does not necessarily improve. But the *actual* condition number of NTK does get improved, as the smallest eigenvalue is improved from zero to non-zero.
> > > Therefore, it is not clear whether the convergence rate can be improved in this case. We will clarify this in the revision.
> > >
> > > The *actual* condition number of NTK (not the “effective” one) is always improved: (1) if $G$ is not full rank, $\kappa$ is improved from infinity to a finite value, since the smallest eigenvalue is improved from zero to non-zero; (2) if $G$ is full rank,  $\kappa$ is also improved, as we discussed in the $d>n$ case. In the latter case, the convergence rate still gets improved by the ReLU activation.
> > >
> > > Although the reviewer might think $d>n$ is not often true in many real problems. We would like to emphasize that our paper is aiming for a theoretical understanding of the activation functions. For this purpose, we believe our paper already provides much clear understandings of the role of ReLU activation, including the *better separation* property, improving the *actual* condition number of NTK, and improving cnovergence rate when $G$ is full rank. Analogously, an infinitely wide network setting, which is often questioned by my practitioners for its non-practical setting, is proven to provide abundant theoretical understandings of neural networks. We hope the reviewer can recognize the value of this paper. Thanks!

---

### Official Review · Reviewer_4jiX · 2023-10-31

**Soundness:** 3 good
**Presentation:** 3 good
**Contribution:** 1 poor
**Rating:** 3
**Confidence:** 4

**Summary:**

This paper studies the effects of ReLU in the neural tangent kernel regime. Specifically, the authors compare ReLU network with linear network and show that (1) ReLU is able to produce better data separation in the feature space of model gradient and (2) ReLU improves the NTK conditioning. The authors further show that depth is able to further amplify those effects.

**Strengths:**

The proof of this work is clean and solid and the presentation of this work is very clear. The idea of analyzing the model gradient feature makes sense and is an interesting subject in kernel learning. I do appreciate the authors' result on providing the exact formula of model gradient angle in Lemma 4.3.

**Weaknesses:**

This work overall gives the reviewer a feeling that it is more or less a direct consequences of [1] as [1] also shows the formula for the angle between post-activations. I applaud the authors for studying Equation (7) and (8) as they are challenging objects. However, the current results (Theorem 4.2 and Theorem 4.4) are only considering the points that are very close to each other $\Theta(x,z) = o(1/L)$ and if $L$ is big, this quantity is very small. As the authors mentioned, for small $z$, $g(z)$ behaves like identity. Thus, although Theorem 4.2 and Theorem 4.4 is able to show that ReLU improves the data separability, the improvement is also very small in the regime the authors are considering in this paper. Thus, the model gradient angles are nearly non-changing from the input angles. This is also why the improvement for the condition number in Theorem 5.2 can also be very small. It would be more interesting to see ReLU can improve data separability for input pair with small angle (but larger than the regime this paper presents). Further, although Proposition 5.1 is able to connect the model gradient angle with the upper bound of the smallest eigenvalue and the lower bound of condition number, it is also not clear how tight the upper bound is.

[1] Arora, Sanjeev, et al. "On exact computation with an infinitely wide neural net." Advances in neural information processing systems 32 (2019).

**Questions:**

Another question that can be explored is whether other non-linear activation has the same properties (better data separation and conditioning) as ReLU. Although for activations other than ReLU, it is significantly more challenging to get a close-form formula.

**Details Of Ethics Concerns:**

None.

---

> ### Author Response · Authors · 2023-11-17
> **Response to Reviewer 4jiX**
>
> We thank the reviewer for the comments. We would like to address your concerns one by one below.
>
> **W1:** *“... it is more or less a direct consequence of [1] as [1] also shows the formula for the angle between post-activation.”*
>
> **A:** The formula for the angle between post-activation is merely the starting point of our analysis. Note that most of our analysis after this formula is very different from that of [1]. We will cite [1] around this formula in the revision.
>
> We would like to draw the reviewer’s attention to our main contribution. The paper showed a new advantage of certain non-linear activations – decreasing the condition number of NTK. Before this, non-linear activations is just known to increase the expressivity. We view this result as an addition to our understanding of the effect of non-linear activation functions. In addition, this new advantage has important implications on the convergence rate.
>
> **W2:** *“the current results (Thm 4.2 and 4.4) are only considering the points that are very close … , if L is big, this quantity is very small.”*
>
> **A:** The small data input angle regime is the key and most interesting part. This is because the NTK condition number is mainly affected by its smallest eigenvalue, which in turn is closely related to the smallest angle between points. A tiny change in the smallest angle between points may lead to a non-negligible change in the NTK smallest eigenvalue $\lambda_{min}$, which may further lead to a significant change in NTK condition number $\kappa$ (as $\kappa \sim 1/\lambda_{min}$, and $\lambda_{min}$ is very small).
>
> **W3:** *“As the authors mentioned, for small $z$, $g(z)$ behaves like identity. … Thus, the model gradient angles are nearly non-changing from the input angles. This is also why the improvement for the condition number in Theorem 5.2 can also be very small. It would be more interesting to see ReLU can improve data separability for input pair with small angle (but larger than the regime this paper presents).” *
>
> **A:** In the small but positive angle regime, a tiny difference between $g(z)$ and $z$ could make a large difference in the NTK condition number. As discussed above, the NTK condition number scales roughly as $1/z$ for very small $z$. Please take a look at the Figure 2(a), for a numerical verification of this better separation property on real-world datasets. It shows that the ReLU non-linearity can significantly increase the smallest angle between points. Moreover, please take a look at Figure 2(b), the condition number improved significantly, often by about $10^2 \sim 10^4$.

---

> > ### Comment · Reviewer_4jiX · 2023-11-22
> > **Response to the Authors**
> >
> > I thank the authors for the clarification.
> >
> > I would like to point out that when I was saying "the improvement on the condition number can be quite small" in my original review, I was referring to theorem 5.2, which shows that depth can improve condition number. Notice that theorem 5.2 only shows $\kappa < \kappa_{0}$ and it is not clear how big the improvement $\kappa_{0} - \kappa$ is. Currently, it seems the improvement is very small. The reason is that the current proof of theorem 5.2 is using theorem 4.4 and theorem 4.4 is only able to show a $o(1)$ improvement in the cosine value of the angles: if we take $\theta_{\text{in}} = o(1/L)$, then equation 9 simplifies to $\cos \phi = (1-o(1)) \cos \theta_{\text{in}}$. If the data set is normalized, then in the context of the proof of theorem 5.2 on page 19, we have $\lambda_2(K), \lambda_2(G) \geq \Omega(1)$. However, $|\lambda_2(K)- \lambda_2(G)| = o(1)$ which only implies a very small improvement on the condition number. Further, the experiment results show that both the angle and the condition number can improve by quite a bit. Thus, if the authors are able to show a result like $\kappa_{0} - \kappa \geq \text{some substantial quantity}$ then I am willing to reconsider my evaluation. However, currently, I remain my previous judgement.

---

> > > ### Author Response · Authors · 2023-11-22
> > >
> > > Thanks for your reply.
> > >
> > > Please note that even such a qualitative result on activation function is not clearly known before. As an opening work in such a direction, we believe a qualitative is alreay significant. Definitely, we agree it is interesting to explore a quantitative result of such kind, we believe it is acceptable to leave as a future work.

---

### Official Review · Reviewer_hbD5 · 2023-11-01

**Soundness:** 3 good
**Presentation:** 2 fair
**Contribution:** 3 good
**Rating:** 6
**Confidence:** 3

**Summary:**

This paper studies the effect of the ReLU activation on the conditioning of the NTK and the separation of data points when passed through a deep neural network. The authors show that in contrast to linear activations, the ReLU activation causes an increase in angle separation of data points as well as an improvement in the condition number of the NTK. Additionally, this effect scales with the depth of the network. They corroborate their theoretical results with numerical experiments on a variety of datasets.

**Strengths:**

1. The main result appears to be fairly novel and interesting.
2. Numerical experiments corroborate the theory well.
3. The experiments are thorough and clear.

**Weaknesses:**

1. The writing style and general clarity in some parts of the paper is lacking.
2. The main theoretical results compare with the baseline of a fully linear network, which is not a very interesting comparison. It does not seem that surprising that a linear model cannot increase angle separation while the ReLU model can. A more interesting comparison would be with other non-linear models like kernel machines for example.
3. The theory of the improved convergence rates of gradient descent is restricted to the infinite or large width regime where the NTK is constant. However, it is observed in practice that the NTK changes significantly in most interesting cases [1], and this change corresponds to important feature learning. Per my understanding, the theory in this work fails to extend to this changing NTK regime.

[1] - Fort, Stanislav, et al. "Deep learning versus kernel learning: an empirical study of loss landscape geometry and the time evolution of the neural tangent kernel." Advances in Neural Information Processing Systems 33 (2020): 5850-5861.

**Questions:**

1. Can the authors comment on possible ways to extend this result to other non-linear models like kernel machines?
2. Does the conditioning result extend to the finite-width empirical NTK?

---

> ### Author Response · Authors · 2023-11-17
> **Response to Reviewer hbD5**
>
> We thank the reviewer for the comments. We would like to address your concerns one by one below.
>
> **W1:** *“The writing style and general clarity in some parts of the paper is lacking”*
>
> **A:** Could you provide more details? For example, which part of the paper is not clear to you.  We are happy to revise wherever was not clear.
>
> **W2:** *“The main theoretical results compare with … a fully linear network … is not a very interesting comparison.”*
>
> **A:** First of all, we would like to restate our main contributions here: the paper showed a new advantage of certain non-linear activations – decreasing the condition number of NTK. (in parallel with the well-known advantage of increasing the expressivity of network functions).
>
> To show this new advantage in decreasing condition number, it is necessary to compare with the *linear network* that has exactly the same architecture except the non-linear activation. It was this comparison that makes this new advantage evident. A comparison with other non-linear models (e.g., kernel machines) does not support such a claim (although we agree that this kind of comparison is an interesting future direction).
>
> Analogously, think about the other well-known advantage of non-linear activation – increasing the expressivity of network function. This better expressivity was made clear by the comparison between non-linear network and linear network, but not between different non-linear activations.
>
> We would like to point out that this paper is never meant to say the ReLU model is superior to any other models. Instead, we aim to deliver a message on understanding the role of non-linear activations.
>
> **W3:** *“It does not seem that surprising that a linear model cannot increase angle separation while the ReLU model can.”*
>
> **A:** We are not aware of any similar results in existing works.
>
> **W4:** *“The theory … is restricted to the infinite or large width regime… in practice the NTK changes significantly …”*
>
> **A:** We would like to point out that many recent theoretical works are conducted in this large width regime, for example [1,2,3,4]. This regime, although many people believe impractical, provides a lot of new understanding of modern deep learning. We believe this is a very good starting point.
>
>
> [1]: Mikhail Belkin, Daniel Hsu, Siyuan Ma, and Soumik Mandal. “Reconciling modern machine learning practice and the classical bias–variance trade-off”. PNAS 2019.
>
> [2]: Zhang, C., Bengio, S., Hardt, M., Recht, B., and Vinyals, O. Understanding deep learning (still) requires rethinking generalization. Communications of the ACM, 64(3):107– 115, 2021.
>
> [3]: Simon Du, Jason Lee, Haochuan Li, Liwei Wang, and Xiyu Zhai. “Gradient Descent Finds Global Minima of Deep Neural Networks”. In: International Conference on Machine Learning. 2019.
>
> [4]: Difan Zou, Yuan Cao, Dongruo Zhou, and Quanquan Gu. “Stochastic gradient descent optimizes over-parameterized deep relu networks”. In: arXiv preprint arXiv:1811.08888 (2018).
>
> [5]: Chaoyue Liu, Libin Zhu, and Mikhail Belkin. “Loss landscapes and optimization in over-parameterized non-linear systems and neural networks”. In: Applied and Computational Harmonic Analysis 59 (2022).

---

> > ### Comment · Reviewer_hbD5 · 2023-11-22
> > **Thank you for your response**
> >
> > Thank you for your detailed clarifications, they are helpful.
> >
> > Indeed, this is a novel viewpoint in understanding the benefits of non-linear activation functions. However, I still believe that the fact that this theory only applies in the infinite width regime is a significant limitation. Can the authors comment on theoretical or experimental evidence that measures the angle-separation property and conditioning of the changing finite-width NTK as training progresses? This would be an interesting observation to further the impact of the results in this paper.
> >
> > Given these points, I am willing to increase my score.

---

> > > ### Author Response · Authors · 2023-11-23
> > > **thanks for your further comments**
> > >
> > > Thanks for your reply.
> > >
> > > First of all, we would like to point out that nearly all optimization theories for neural networks so far are either in the infinite width limit or with super large width (or more precisely, require sufficiently large network width) (see, e.g., [1-5]). To the best of our knowledge, there is no general theory for convergence of GD/SGD outside of large width regimes. The main technique in those works is the infinitesimal or sufficiently small change of NTK during training.
> > >
> > > Using the same technique of small NTK change, Theorem 5.2 (NTK for deep networks with two data samples) can be easily extended to the setting of finite but large width (as can be seen from its proof, the eigenvalues have closed form expressions, and can be quantatively analyzed). As for Theorem 5.3, in order to extend to finite width, we need a bit of extra analysis that can quantitatively control the difference of NTKs between with and without ReLU. But as the NTK change is small, we believe it is not a hard extension.
> > >
> > > Indeed, we don’t view the infinite width limit as a significant limitation. Please see the original NTK paper [6], as well as [7] for example. The analyses of these works are conducted in this infinite width limit. It turns out that these results have brought significant insights to neural networks.
> > >
> > >
> > >
> > > [1] Allen-Zhu, Li, Song. A Convergence Theory for Deep Learning via Over-Parameterization. ICML. 2019.
> > >
> > > [2] Du, Zhai, Poczos, Singh. Gradient Descent Provably Optimizes Over-parameterized Neural Networks. ICLR, 2018.
> > >
> > > [3] Zou, Cao, Zhou, Gu. Gradient descent optimizes overparameterized deep ReLU networks. Machine Learning, 2020.
> > >
> > > [4] Oymak, Soltanolkotabi. Toward moderate overparameterization: Global convergence guarantees for training shallow neural networks. : IEEE Journal on Selected Areas in Information Theory, 2020.
> > >
> > > [5] Liu, Zhu, Belkin, Loss landscapes and optimization in over-parameterized non-linear systems and neural networks. ACHA, 2022.
> > >
> > >
> > > [6] Jacot, Gabriel, Hongler. “Neural tangent kernel: Convergence and generalization in neural networks”. NeurIPS. 2018.
> > >
> > >
> > > [7] Lee, Xiao, Schoenholz, Bahri, Novak, SohlDickstein, Pennington. “Wide neural networks of any depth evolve as linear models under gradient descent”. NeurIPS 2019

---

### Official Review · Reviewer_WJEr · 2023-11-12

**Soundness:** 3 good
**Presentation:** 3 good
**Contribution:** 2 fair
**Rating:** 5
**Confidence:** 4

**Summary:**

This paper studies several properties of wide neural networks in the neural tangent kernel (NTK) regime. By comparing the cases with and without the ReLU activation function, it is shown that ReLU has the effects of (i) better data separation, and (ii) better NTK conditioning. These results also indicate that deeper ReLU networks have better effects, and that ReLU activations can accelerate the optimization procedure.

**Strengths:**

- This paper introduces an interesting perspective in the study of deep neural networks focusing on the angles between data in the feature space.
- The presentation of the paper is clear.
- The experiments match theory results well.

**Weaknesses:**

- The major conclusions of this paper (about the advantages of ReLU) are only demonstrated in comparison with linear networks. This makes the results not very strong, as the advantages of non-linear activations over linear networks are fairly clear.
- Since the comparisons are only made between ReLU networks and linear networks, the results of the paper may not be very comprehensive. For example, in the title, “ReLU soothes NTK conditioning” may give readers the impression that ReLU activation has a unique property when compared with other activation functions, which is not really the case. The results would be more comprehensive if the authors can extend the comparison between ReLU and linear activation functions to a comparison between a general class of non-linear activations and the linear activation.
- The results of this paper may not be very surprising. As the authors mentioned, the major known advantage of non-linear activation is to improve the expressivity of neural networks. It seems that the conclusions of this paper, to a large extent, are still saying the same thing. Better data separation, better NTK conditioning, and faster convergence to zero training loss, all seem to be more detailed descriptions of better expressivity.
- The impact of the results are not sufficiently demonstrated. For example, it is not very clear what is the benefit to achieve better data separation in the feature space.

**Questions:**

I suggest that the authors should consider address the comments in the weaknesses section.

---

> ### Author Response · Authors · 2023-11-17
> **Response to Reviewer WJEr**
>
> We thank the reviewer for the comments. After carefully reading the review, we think the reviewer’s concerns are largely based on a misunderstanding of the paper. We apologize for not making it crystal clear in the submission. We hope the following explanation can address all the concerns.
>
> **W1:** *“The major conclusions of this paper ……,as the advantages of non-linear activations over linear networks are fairly clear”.*
>
> **A:** We politely could not fully agree with this point. The advantage of non-linear activations was only **partially** clear. It was only known that non-linear activations increase the expressivity of the network function (i.e, can approximate more complicated functions). However, our paper showcases a new advantage: decreasing the NTK condition number (which in turn induces a faster worst-case convergence rate). **This advantage was not known before.**
>
> **W2:** *“The comparisons are only made between ReLU and linear networks …”*
>
> **A:** As we clarified above, our major contribution is to showcase an never-noticed advantage of certain non-linear activations.
>
> Let’s make an analogy. Think about the advantage of non-linear activation in increasing expressivity. When talking about this advantage, it was the comparison between non-linear network and linear network that makes this advantage clear. Comparing between different activations does not support this known advantage of increasing expressivity.
>
> Same thing here. To show this new advantage of decreasing condition number, it is necessary to compare with the *linear network* that has exactly the same architecture except the non-linear activation. Comparing different activations only implies which activation might be relatively better or worse (we agree this is an interesting point, but not the main scope of this paper).
>
> **W3:**  *“The results of this paper may not be very surprising. … the conclusions of this paper are still saying the same thing. .. all seem to be more detailed descriptions of better expressivity”.*
>
> **A:** We could not agree with this point, either. This new advantage of certain non-linear activation (i.e., better separation, decreasing NTK etc) is absolutely different/independent from the better expressivity. We don’t see any connections between the two, and are not aware of any existing result making this connection. If you think the two advantages are the same, please provide details/evidence. We are happy to discuss this point.
>
> **W4:**  *“It is not very clear what is the benefit to achieve better data separation in the feature space.”*
>
> **A:** As we elaborated in the paper, the better separation is deeply related to the better NTK conditioning, as well as the faster worst-case convergence rate. Loosely speaking, it is the better separation that leads to the better NTK conditioning. Intuitively, a better data separation in the feature space helps the learning of the model. Think about two similar data points with different labels, which is often hard to be distinguished by a model. With a better data separation in the feature space, it becomes easier for the model to distinguish. We will add a discussion of the intuition in the revision.
>
>
> Overall, we think the newly discovered advantage of certain non-linear activation (i.e., better separation, better NTK conditioning, etc) is important for theoretical understanding of the neural network, and should be inspiring for further research. We hope the reviewer can recognize the novelty and importance of this discovery.

---

### Meta-Review · Area_Chair_7hay · 2023-12-06

**Metareview:**

This paper reveals the impact of the ReLU activation function, demonstrating two key effects: (a) enhanced data separation, indicated by a larger angle separation for similar data in the feature space of model gradient, and (b) improved NTK conditioning, reflected in a smaller condition number of the neural tangent kernel (NTK). Additionally, the study highlights that increasing the ReLU network depth further amplifies these effects. However, the reviewers express concerns, including (1) the perceived ease of obtaining the results from existing work; and (2) the restrictive settings, such as focusing on two data points, NTK regime, and treating sample size and dimension as constants. Despite author responses and discussions, the paper does not get sufficient support. So I have to recommend rejection.

**Justification For Why Not Higher Score:**

This paper suffers several weaknesses. Despite author responses and discussions, the paper does not get sufficient support.

**Justification For Why Not Lower Score:**

N/A

---

### Decision · Program_Chairs · 2024-01-16

Reject